# Adeno-associated virus-vectored influenza vaccine elicits neutralizing and Fcγ receptor-activating antibodies

Daniel E Demminger[1], Lisa Walz[2], Kristina Dietert[3], Helen Hoffmann[4], Oliver Planz[4], Achim D Gruber[3], Veronika von Messling[2] & Thorsten Wolff[1,*]

## Abstract

The current seasonal inactivated influenza vaccine protects only against a narrow range of virus strains as it triggers a dominant antibody response toward the hypervariable hemagglutinin (HA) head region. The discovery of rare broadly protective antibodies against conserved regions in influenza virus proteins has propelled research on distinct antigens and delivery methods to efficiently induce broad immunity toward drifted or shifted virus strains. Here, we report that adeno-associated virus (AAV) vectors expressing influenza virus HA or chimeric HA protected mice against homologous and heterologous virus challenges. Unexpectedly, immunization even with wild-type HA induced antibodies recognizing the HA-stalk and activating FcγR-dependent responses indicating that AAV-vectored expression balances HA head- and HA stalk-specific humoral responses. Immunization with AAV-HA partially protected also ferrets against a harsh virus challenge. Results from this study provide a rationale for further clinical development of AAV vectors as influenza vaccine platform, which could benefit from their approved use in human gene therapy.

**Keywords** adeno-associated virus vector; broadly reactive antibody; FcγR; HA stalk; universal influenza vaccine

**Subject Categories** Immunology; Microbiology, Virology & Host Pathogen Interaction

See also: **JA Bengoechea** (May 2020)

## Introduction

Influenza remains a severe public health threat. The infection is associated with high morbidity and mortality, especially in very young or very old individuals, and has thus considerable socio-economic impact (WHO, 2018b). Currently, influenza A viruses of the subtypes H1N1 and H3N2 as well as the influenza B virus lineages Victoria and Yamagata circulate in humans. Influenza viruses are genetically and antigenically highly variable, resulting in recurrent epidemics in humans ("flu season"). The main mechanism driving this variability is antigen drift caused by the accumulation of point mutations in the antigenic surface glycoproteins hemagglutinin (HA) and neuraminidase (NA). Moreover, reassortment between influenza A virus subtypes can generate even larger antigenic alterations, which may allow for a pandemic circulation in a naïve population (Palese, 2004). Influenza pandemics occur in unpredictable intervals. In 1918, the most devastating pandemic caused by an H1N1 virus, the "Spanish flu," claimed millions of deaths worldwide (Johnson & Mueller, 2002). The most recent influenza pandemic was triggered by a related H1N1 virus in 2009, which arose from the porcine reservoir and imposed a high burden of disease on public health (Fineberg, 2014).

Currently, the most effective prophylaxis against influenza is immunization with trivalent or quadrivalent influenza vaccines, most of which contain inactivated antigenic components of current seasonal influenza A and B viruses. However, protection is mainly virus strain-specific, and the efficacy against non-matched strains is generally poor. Although a more broadly reactive live-attenuated influenza vaccine (LAIV) is available for children, its usefulness is limited in adults due to pre-existing immunity against previously encountered influenza viruses (Belshe *et al*, 2000). The majority of vaccine doses is produced in embryonated chicken eggs, a process which is time-consuming and necessitates that the vaccine composition is predictively defined well in advance of the seasonal epidemic (Gerdil, 2003). This elevates the risk for vaccine mismatch and loss of effectiveness if the actually circulating influenza viruses represent drift variants of the vaccine strains (Rondy *et al*, 2017). Seasonal influenza vaccination also does not provide protection against shifted or emerging zoonotic influenza A virus strains, e.g., H5N1 or H7N9, which can be associated with increased disease severity

1 Unit 17—Influenza and Other Respiratory Viruses, Robert Koch Institute, Berlin, Germany
2 Veterinary Medicine Division, Paul-Ehrlich-Institute, Langen, Germany
3 Department of Veterinary Medicine, Institute of Veterinary Pathology, Berlin, Germany
4 Department of Immunology, Interfaculty Institute for Cell Biology, Eberhard Karls University, Tübingen, Germany
*Corresponding author. Tel: +49 30 187542278; E-mail: WolffT@rki.de

(WHO, 2018a). Hence, there is a generally accepted impetus for development of a novel broadly protective vaccine (Erbelding *et al*, 2018; Ortiz *et al*, 2018).

The seasonal vaccine predominantly induces antibodies targeting epitopes at or surrounding the receptor binding site (RBS) within the globular HA-head region (Caton *et al*, 1982). Since these epitopes are highly variable, antibodies show limited cross-protection (Yu *et al*, 2008). Significantly, broadly protective antibodies recognizing conserved epitopes in the membrane proximal HA-stalk or in the HA-head region have been discovered in mice and humans recently (Wu & Wilson, 2017). The majority of broadly reactive HA antibodies does not interfere with receptor attachment (Brandenburg *et al*, 2013). They rather execute their protective effect *via* interference with later steps in the viral replicative cycle or *via* Fc-receptor (FcR)-mediated mechanisms, including antibody-dependent cellular cytotoxicity (ADCC) (DiLillo *et al*, 2014). Vaccination with inactivated influenza antigen alone is not able to efficiently induce ADCC-activating antibodies in a non-human primate model (Jegaskanda *et al*, 2013). However, pre-existing ADCC-activating antibody titers induced following natural infection could be boosted in humans by inactivated vaccine, indicating that efficient priming is imperative to elicit such antibodies (Jegaskanda *et al*, 2016).

Overall, the level of broadly reactive antibodies in infected or vaccinated individuals is low due to immunodominance of variable epitopes in the HA-head region (Ellebedy *et al*, 2014). However, substantial antigenic changes of the HA-head region, for example as consequence of antigenic shift, can lead to the expansion of HA-stalk antibodies, in which the stalk-specific recall response outcompetes the *de novo* response against the shifted head (Li *et al*, 2012). Immunization strategies employing chimeric HA (cHA) or headless HA thus make use of this phenomenon (Steel *et al*, 2010; Hai *et al*, 2012; Li *et al*, 2012; Impagliazzo *et al*, 2015; Yassine *et al*, 2015). However, since the production of these antigens either includes vaccine bulk production in eggs or requires technically challenging protein purification processes as well as adjuvants, alternative vaccine platforms offer an attractive development perspective (Ramezanpour *et al*, 2016; Grimm & Buning, 2017). Vaccinia- or adenovirus-based vectors currently represent the most widely used platforms in clinical vaccine trials (Ramezanpour *et al*, 2016). However, adeno-associated virus (AAV) vectors might be particularly suited as influenza vaccine carrier, since AAV is naturally replication-incompetent and apathogenic in humans, which was a prerequisite for licensure as the first gene therapy vector for use in humans (Grimm & Buning, 2017). The "gutless" AAV vectors can be produced readily in cell culture according to good manufacturing practice criteria, avoiding some of the aforementioned limitations regarding vaccine production (Tripp & Tompkins, 2014). Also, AAV vectors are re-administrable into the respiratory tract in the context of pre-existing immunity without need to change the vector capsid (Limberis & Wilson, 2006). Intriguingly, AAV vectors have been used for passive immunization of mice and ferrets via expression of broadly reactive HA-stalk antibodies in the respiratory tract (Balazs *et al*, 2013; Limberis *et al*, 2013a,b; Adam *et al*, 2014; Laursen *et al*, 2018). Furthermore, active vaccination with AAV vectors expressing internal (nucleoprotein (NP), matrix protein 1) and surface (HA) influenza virus antigens protected mice from challenge infection (Xin *et al*, 2001; Lin *et al*, 2009; Sipo *et al*, 2011). In the absence of neutralizing antibodies against heterologous HA, the protection

against a non-matched influenza virus observed in that study was attributed to the presence of cross-reactive T cells. However, the quality and influence on protection of non-neutralizing antibodies were neither evaluated nor compared to an inactivated vaccine (Sipo *et al*, 2011). Furthermore, immunization with AAV-vectored cHA or headless HA antigens has not been tested, and there are no data as to the transferability of active AAV vector vaccination to the ferret model, thought to most accurately represent human influenza disease (Enkirch & von Messling, 2015). A detailed knowledge of the effects of the AAV vector on the immune response is needed to advance the AAV vector vaccine approach toward clinical development (de Vries & Rimmelzwaan, 2016).

Here, we show that vaccination of mice with AAV-HA or AAV-cHA induced broadly protective antibodies. Notably, protection was associated with strong induction of FcγR-activating antibodies. Finally, we were able to show that three doses of an AAV-HA vaccine conferred protection in ferrets against an (H1N1)pdm strain, demonstrating the potential of the platform for further development.

# Results

### AAV9-vectors induce strong antigen expression *in vitro*

We evaluated the potency of AAV vectors expressing influenza virus wild-type HA and NP, or cHA and headless HA antigens to confer broad protection from influenza virus challenge in comparison with an inactivated vaccine (Fig 1A). The AAV9 serotype used herein was isolated from human tissue and efficiently transduces respiratory tissue of mice and ferrets (Gao *et al*, 2004; Limberis & Wilson, 2006; Limberis *et al*, 2013a). All vaccine constructs were based on proteins encoded by the prototypic pandemic influenza virus A/California/7/2009 (H1N1)pdm (Cal/7/9) (Fig 1B). The cHA contained the stalk region of Cal/7/9 and head regions derived from influenza A virus subtypes H2 (cHA1), H10 (cHA2), or H13 (cHA3) that currently not circulate in humans. Furthermore, three different headless HA constructs were tested, either containing merely the deletion of the head region (headless HA, HL), or further modifications that increase their antigenicity (modified headless 1 and 2, mHL1 and mHL2; Fig 1B; Impagliazzo *et al*, 2015; Steel *et al*, 2010; Yassine *et al*, 2015). Initially, we assessed correct folding of the HA-stalk within the constructs with the prototypic conformational stalk antibody C179, as several broadly reactive antibodies recognize conformational epitopes (Fig 1C; Okuno *et al*, 1993). Wild-type HA and cHA3 displayed the C179 epitope, while cHA1 and mHL1 showed reduced binding to C179 (Fig 1C). cHA2, headless HA and mHL2 were only faintly detectable with C179 (Fig 1C). Of note, all cHA were detected by immune sera raised against the respective parental influenza virus HA subtype, i.e., H2, H10, or H13, indicating correct recapitulation of the structure of the respective HA head (Appendix Fig S1A and B). The order of the cHA for vaccination of animals was set according to C179-staining intensity, i.e., cHA3–cHA1–cHA2 (Appendix Table S1).

AAV2-vectors transencapsidated into AAV9 capsids were produced, and integrity and identity of the AAV9 capsids were verified using the conformational capsid antibody ADK9 (Appendix Fig S1C and D; Sonntag *et al*, 2011). All vector stocks contained high amounts of encapsidated viral genomes (vg) and relatively

low amounts of empty capsids (Appendix Fig S1E and F). Correspondingly, robust *in vitro* AAV vector transduction rates were achieved (Fig 1D and Appendix Fig S1G).

## AAV-HA, AAV-cHA, AAV-NP, and inactivated vaccine induced broadly reactive antibodies in mice

To assess immunogenicity of the AAV vector vaccines, 50 μl PBS containing $10^{11}$ vg per mouse was applied equally to both nostrils three times in 3-week intervals before being challenged with influenza viruses. Control groups received either three times AAV-GFP or two times Cal/7/9 whole-inactivated virus (WIV) via the same route in order to be consistent with the application of the AAV-vector vaccines (Fig 2A, Appendix Table S1). Earlier analysis had shown that intranasally applied WIV vaccine elicits protective anti-influenza immune responses in mice (Bhide *et al*, 2019). As expected, all animals vaccinated with AAV vectors developed high anti-AAV9 IgG titers which continued to increase over time (Fig EV1A). Furthermore, AAV9-vector-neutralizing antibodies were induced, which correlated with total anti-AAV9 serum IgG titers (Fig EV1B and C).

The reactivity breadth of serum antibodies was tested against a panel of ten influenza viruses from both antigenic group 1 (H1N1, H2N3, H5N1, H13N6) and 2 (H3N2, H7N9, H10N7; Fig 2).

Immunization with AAV-HA, AAV-cHA, AAV-NP, and WIV induced a significant increase of homologous Cal/7/9-specific serum IgG antibody titers compared to pre-immune serum titers (Fig 2B). Furthermore, AAV-HA- and AAV-NP-immunized mice had significantly higher titers against Cal/7/9 compared to AAV-cHA or WIV (Fig 2B). Interestingly, none of the AAV-vectored headless HA vaccines induced detectable influenza-specific antibodies or antibodies against the respective AAV-vectored antigen (Fig EV1D–F). This might be due to the lack of immunodominant epitopes in these antigens. Based on a report by Hessel *et al* (2014), we evaluated whether the combination of AAV-mHL with the highly immunogenic AAV-NP would induce HA-stalk antibodies. This, however, was not the case, and only NP reactive antibodies were induced (Fig EV1G). Groups receiving AAV-vectored headless HA were therefore not included in subsequent analyses.

AAV-HA, AAV-cHA, AAV-NP, and WIV induced broadened antibody responses (Fig 2D–G). AAV-HA triggered a strong response mainly against H1N1 viruses, including pandemic H1N1 virus from 1918, but also H5N1 (Fig 2D). Although reacting weaker with Cal/7/9 and the 1918 pandemic H1N1 viruses, AAV-cHA sera reacted also with H5N1 and two of the cHA parental group 1 viruses (subtypes H2 and H13) (Fig 2E). Both, AAV-HA and AAV-cHA, did, however, not induce antibodies against group 2 viruses (Fig 2D and

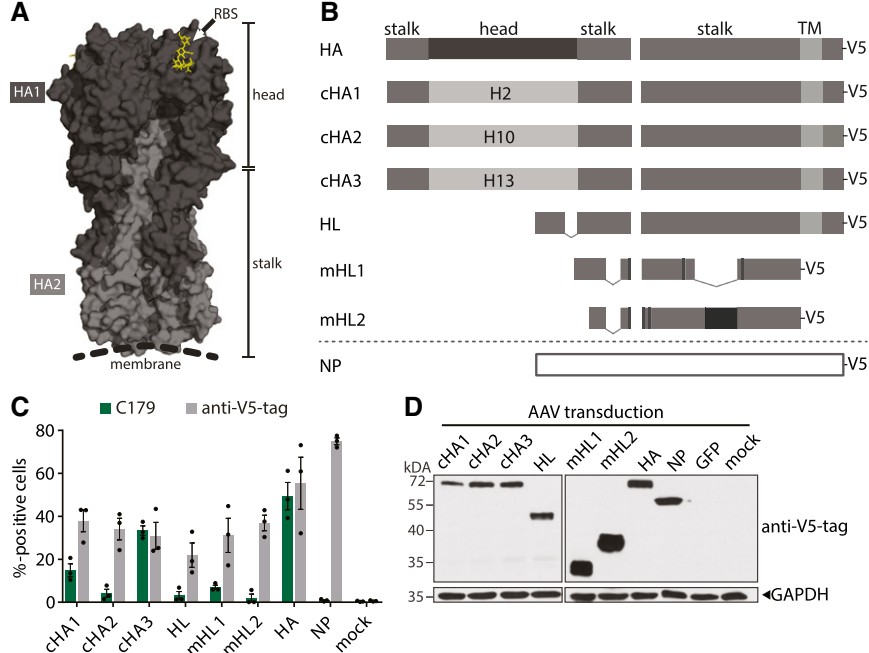

**Figure 1.  (H1N1)pdm-based AAV-vectored antigens are strongly expressed *in vitro*.**

A   3D structure of a HA trimer (PDB 3UBE generated with PyMol). Each monomer consists of a HA1 (dark gray) and HA2 (light gray) subunit. The trimer can be divided into a membrane distal head which contains the RBS (yellow) and a proximal stalk region.

B   HA and NP represent the Cal/7/9 wild-type proteins. Chimeric HA (cHA) 1 contains the head regions of H2 HA, cHA2 of H10 HA, and cHA3 of H13 HA, while they all contain the Cal/7/9 HA-stalk region. Headless HA (HL) contains a deletion in the HA-head region (dashed line). Modified headless HA (mHL1 and mHL2) contain stabilizing mutations (black boxes) and lack additional internal parts. All constructs were codon-optimized and carry a V5-tag at their C-terminus.

C   Frequency of C179$^+$ or V5-tag$^+$ 293T cells 24 h after transfection of AAV vector plasmids as measured by flow cytometry. Symbols represent single experiments and bars the mean ± SE (*n* = 3).

D   Immunoblot of 293T cells 72 h after transduction with AAV vectors at a MOI of $10^6$. Antigen expression was detected with an anti-V5-tag antibody. Equal loading was controlled with a GAPDH antibody (*n* = 3).

Source data are available online for this figure.

E). In contrast, AAV-NP induced a strong antibody response covering viruses from both antigenic HA groups, including subtypes H3N2 and H7N9, most likely due to the high conservation of NP (Fig 2F). Unexpectedly, WIV vaccination also induced broadly reactive antibodies covering several subtypes of group 1 and 2, though at lower intensities (Fig 2G).

IgA antibodies confer protection to respiratory pathogens due to their high local abundance in the airway mucosa (Asahi *et al*, 2002). We determined pre-challenge levels of serum IgA (Fig EV1H) as well as post-challenge levels of IgA in lung homogenates (Figs 2C and EV1I). AAV-HA and AAV-NP immunization induced IgA antibodies in the serum as well as in the lung against Cal/7/9. In contrast, IgA was not or barely detectable in serum and lung, respectively, after AAV-cHA immunization. Only AAV-NP-immunized mice mounted serum and lung IgA antibodies against heterologous A/Puerto Rico/8/1934 (H1N1) (PR8) virus (Fig EV1H and I). Interestingly, increased levels of IgA were detectable in lung homogenates but not in sera of WIV-immunized mice against both Cal/7/9 and PR8 (Figs 2C, and EV1H and I). In summary, antibodies were

induced at least against some group 1 HA including the subtypes H1N1 and H5N1 after immunization with AAV-HA or AAV-cHA, while AAV-NP and WIV immunization led to antibody responses reacting with influenza A viruses from both antigenic groups.

### Broadly reactive HA-specific antibodies are non-neutralizing *in vitro*

To analyze the characteristics of the serum antibody responses, we initially determined hemagglutination inhibition (HAI) and neutralizing antibody titers. The level of $HAI^+$ antibodies that block the RBS and interfere with attachment is a key parameter for evaluation of currently licensed inactivated vaccines. Neutralizing antibodies, though not necessarily binding directly to the RBS, can also inhibit later steps in the viral replication cycle as well (Brandenburg *et al*, 2013). To capture such effects, we performed microneutralization (MN) assays (He *et al*, 2015). $HAI^+$ and $MN^+$ antibodies against the homologous Cal/7/9 virus were found only in animals immunized with AAV-HA (Table 1). However, these antibodies were

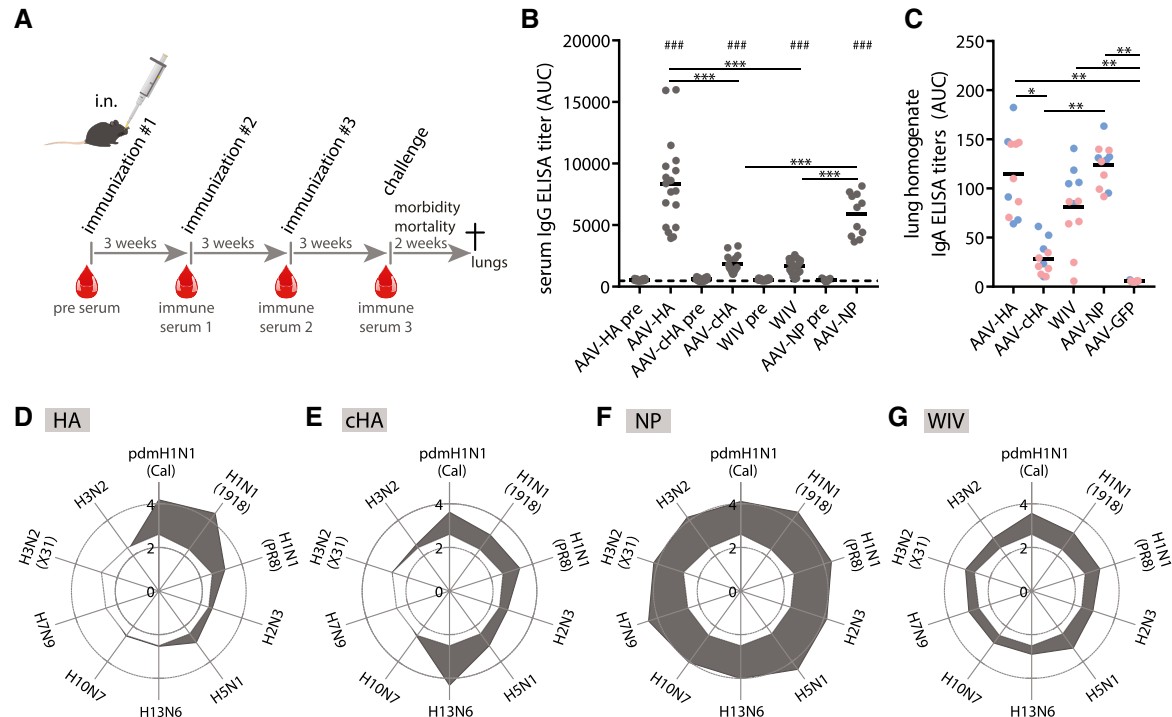

**Figure 2. AAV-vectored vaccines and WIV induce broadly reactive antibodies.**

A　Mice were intranasally given $10^{11}$ vg of AAV vector in 50 µl volume three times in 3-week intervals. 20 µg of Cal/7/9 WIV per 50 µl was given i.n. two times. Blood samples were taken at indicated time points (red drops). After influenza challenge, mice were monitored for survival and weight loss for 2 weeks before necropsy.

B　Total IgG ELISA titers expressed as area under the curve (AUC) against homologous Cal/7/9 virus of pre-immune and pre-challenge sera of individual animals of the indicated vaccine groups (AAV-HA, -cHA, -GFP, WIV $n = 18$, AAV-NP $n = 11$). Background reactivity of pooled AAV-GFP pre-challenge serum is shown as dashed line. ELISAs were performed in technical duplicates.

C　IgA ELISA titers in post-challenge lung homogenates against Cal/7/9 of individual mice of the indicated vaccine groups (AAV-HA, -cHA, -GFP, -NP, WIV $n = 11$). of the Cal/7/9 (blue symbols) and PR8 low-dose (red symbols) challenge groups. ELISAs were performed in technical duplicates.

D–G　Total IgG ELISA titers expressed as $\log_{10}$ of the mean AUC against indicated viruses in pre-challenge sera in AAV-HA (D, $n = 18$)-, AAV-cHA (E, $n = 18$)-, AAV-NP (F, $n = 11$)-, or WIV (G, $n = 18$)-immunized animals (gray area). Reactivity of sera of AAV-GFP-immunized animals is indicated as white area in the center of each web diagram. ELISAs were performed in technical duplicates.

Data information: Statistical significance between pre-immune and pre-challenge serum was determined using Wilcoxon matched pairs test ($^{###}P < 0.001$). Statistical significance between vaccine groups was determined using Kruskal–Wallis test with Dunn's multiple comparison testing ($*P < 0.05$, $**P < 0.01$, $***P < 0.001$). Lines indicate mean.

specific for Cal/7/9 and did not react with another H1N1 or H3N2 virus. Sera from AAV-cHA-immunized animals were $MN^+$ and $HAI^+$ against H13N6, i.e., the parental subtype of cHA3 used for prime immunization, but were negative for all other tested viruses including the parental subtypes of the cHA used for immunization 2 and 3 (H2, H10) (Appendix Table S1, Table 1). Intranasal WIV, as well as AAV-NP and AAV-GFP did not induce $HAI^+$ or $MN^+$ antibodies (Table 1). The lack of $MN^+$ and $HAI^+$ antibodies following WIV application depended on the intranasal immunization route, as intramuscular (i.m.) injection with the same vaccine preparation led to the expected robust induction of $MN^+$ and $HAI^+$ antibodies (Appendix Fig S2). These results indicate that immunization by AAV in our scheme elicited $MN^+$ and $HAI^+$ antibodies toward the HA-head domain of the virus used for prime immunization.

## AAV-HA, AAV-cHA, and WIV induced distinct HA-specific antibody profiles

Recent work on humoral responses against HA has raised great interest into non-neutralizing antibodies that bind outside the canonical antigenic sites, but are capable of activating ADCC or other protective responses (Henry Dunand *et al*, 2016; Leon *et al*, 2016; Tan *et al*, 2016; Wu & Wilson, 2017). Hence, we investigated in more detail the epitopes of HA-specific antibodies in the mouse sera. We first analyzed *via* immunoblot the differential binding to HA1 and HA2 subunits of four different H1N1 viruses spanning more than 90 years of influenza virus evolution (Fig EV2A). HA1 contains the head region, whereas most of the stalk is located on HA2. All serum pools were diluted equally allowing to compare the relative abundances of antibodies recognizing either HA1 or HA2, respectively, among the vaccine groups. AAV-HA vaccination induced antibodies reacting with the HA1 domain of the pandemic Cal/7/9 and A/Brevig Mission/1/1918 (BM/1/1918) viruses, but not of PR8 or seasonal A/Brisbane/59/2007 (Bris/59/7) virus (Figs 3A and EV2B). This is in line with the related antigenicity of the two pandemic strains (Medina *et al*, 2010; Fig EV2A). AAV-HA serum also detected the HA2 of the Cal/7/9, PR8, and Bris/59/7 strains, indicating the additional induction of antibodies binding to the HA stalk (Figs 3A and EV2B). The induction of stalk antibodies was also suggested by reactivity of AAV-HA serum toward headless and chimeric HAs that were in parallel confirmed to carry a native stalk conformation as judged by C179 binding (Fig EV2C). Compared to that, AAV-cHA immunization induced a stronger HA2 antibody response recognizing all four H1N1 viruses, but no antibodies against HA1 (Fig 3A). Interestingly, the difference in reactivity between AAV-HA and AAV-cHA induced antibodies toward a natively folded H1-stalk-antigen as measured by in-cell ELISA was not as pronounced as toward the denatured antigen in immunoblot, indicating that AAV-HA sera contain a considerable proportion of conformational HA-stalk antibodies (Fig EV2C). An epitope screen with overlapping 15-mer peptides derived from Cal/7/9 HA verified the presence of HA-head and HA-stalk antibodies in the AAV-HA sera as it identified five surface-exposed peptides including amino acid positions 168–182 (Pep16), 308–322 (Pep30), 398–412 (Pep39), 418–432 (Pep41), and 488–502 (Pep48) reacting significantly stronger with AAV-HA serum compared to AAV-GFP negative control serum (Fig EV2D). Mapping of these peptides onto the 3D-structure of H1N1 HA revealed that the identified epitopes included

**Table 1. HAI and $MN_{50}$ antibody titers (mouse study).**

| | AAV-HA | AAV-cHA | WIV | AAV-NP | AAV-GFP | Pre |
|---|---|---|---|---|---|---|
| (H1N1)pdm (A/Cal/7/9) | | | | | | |
| HAI | 640 | < 40 | < 40 | < 40 | < 40 | < 40 |
| $MN_{50}$ | 2,794 | < 40 | < 40 | < 40 | < 40 | < 40 |
| H1N1 (A/PR/8/34) | | | | | | |
| HAI | < 40 | < 40 | < 40 | < 40 | < 40 | < 40 |
| $MN_{50}$ | < 40 | < 40 | < 40 | < 40 | < 40 | < 40 |
| H3N2 (A/X31) | | | | | | |
| HAI | < 40 | < 40 | < 40 | < 40 | < 40 | < 40 |
| $MN_{50}$ | < 40 | < 40 | < 40 | < 40 | < 40 | < 40 |
| H13N6 | | | | | | |
| HAI | < 40 | 160 | < 40 | < 40 | < 40 | < 40 |
| $MN_{50}$ | < 40 | 1,047 | < 40 | < 40 | < 40 | < 40 |
| H2N3 | | | | | | |
| HAI | < 40 | < 40 | < 40 | < 40 | < 40 | < 40 |
| H10N7 | | | | | | |
| HAI | < 40 | < 40 | < 40 | < 40 | < 40 | < 40 |

amino acids surrounding the RBS, at the lateral site of the head and/or the upper part of the stalk region, as well as at the membrane proximal part of the HA stalk (Fig 3B). Interestingly, and quite unexpected, intranasal WIV consistently induced mostly HA2-specific antibodies, too (Fig 3A). When given by the i.m route, however, the WIV preparation elicited the expected spectrum of antibodies binding also to HA1 and containing $HAI^+$ and $MN^+$ activity confirming its immunogenicity (Appendix Fig S2). No peptide with significant binding was identified in the epitope screen with intranasal WIV or AAV-cHA antisera, although a peptide with increased reactivity was identified with AAV-cHA sera, which appeared to be buried within the HA stalk (Fig EV2E (arrow) and F).

Upon acidification of the endosome HA undergoes major conformational changes, which eventually results in fusion of viral and endosomal membranes and release of the viral genome into the cytoplasm. HA-stalk binding antibodies may interfere with these steps and thus execute an inhibitory effect on viral replication (Ekiert *et al*, 2011). However, antibodies preventing HA conformational changes were shown not to bind to the low-pH conformation of HA (Friesen *et al*, 2014). Therefore, we analyzed by ELISA the binding of serum antibodies to HA after pre-incubation at neutral to acidic pH and after the removal of the HA1 subunit by treatment with the reducing agent DTT at acidic pH. As expected, binding of the conformational antibody C179 was strongly reduced at low pH against Cal/7/9 and PR8 (Fig 3C). Sera induced by AAV-HA bound to Cal/7/9 unless the HA1 subdomain was removed, indicating that homologous binding was mediated mainly by antibodies against the HA head (Fig 3D). This was, however, not the case for PR8, to which binding was presumably mediated by HA-stalk antibodies (Fig 3A and D). Interestingly, unlike C179, AAV-HA and AAV-cHA induced sera-contained antibodies that bind to the low-pH conformation of the HA stalk (Fig 3D and E). The even stronger binding observed with AAV-cHA serum might result from exposition of

epitopes after pH-induced conformational change, which are otherwise buried within the HA stalk (Fig EV2E). Binding of WIV-induced antibodies did not vary at different pH or with removal of HA1 (Fig 3F), which was also the case for AAV-NP sera (Fig EV2G). In conclusion, AAV-HA induced antibodies against both the HA head and stalk, while AAV-cHA and WIV elicited mainly HA-stalk reactive antibodies.

### AAV-vectored vaccines induced broadly reactive FcγR-activating antibodies

To address whether AAV vector vaccination induced broadly reactive antibodies with the ability to activate FcγR and thereby upregulate

antiviral effector mechanisms, we used an assay that allows for separate analyses of all four murine FcγR (FcγRI, FcγRIIB, FcγRIII, and FcγRIV) (Fig EV3A; Van den Hoecke *et al*, 2017). AAV-HA, AAV-NP, and AAV-cHA vaccines clearly more potently induced Cal/7/9 virus-specific FcγR-activating antibodies than WIV (Fig 4A). Moreover, only the AAV-vectored vaccines but not WIV induced FcγR-activating antibodies against a heterologous virus strain, i.e., PR8, although at lower levels (Fig 4B). Notably, AAV-HA, AAV-cHA, and WIV total PR8 antibody titers were not predictive for FcγR antibody titers (Fig EV3B). Interestingly, AAV-NP sera induced strong FcγR-activating antibody responses against both viruses (Fig 4A and B).

While it was initially proposed that only HA-stalk antibodies rely on activation of FcγR to execute their protective effect, this view has

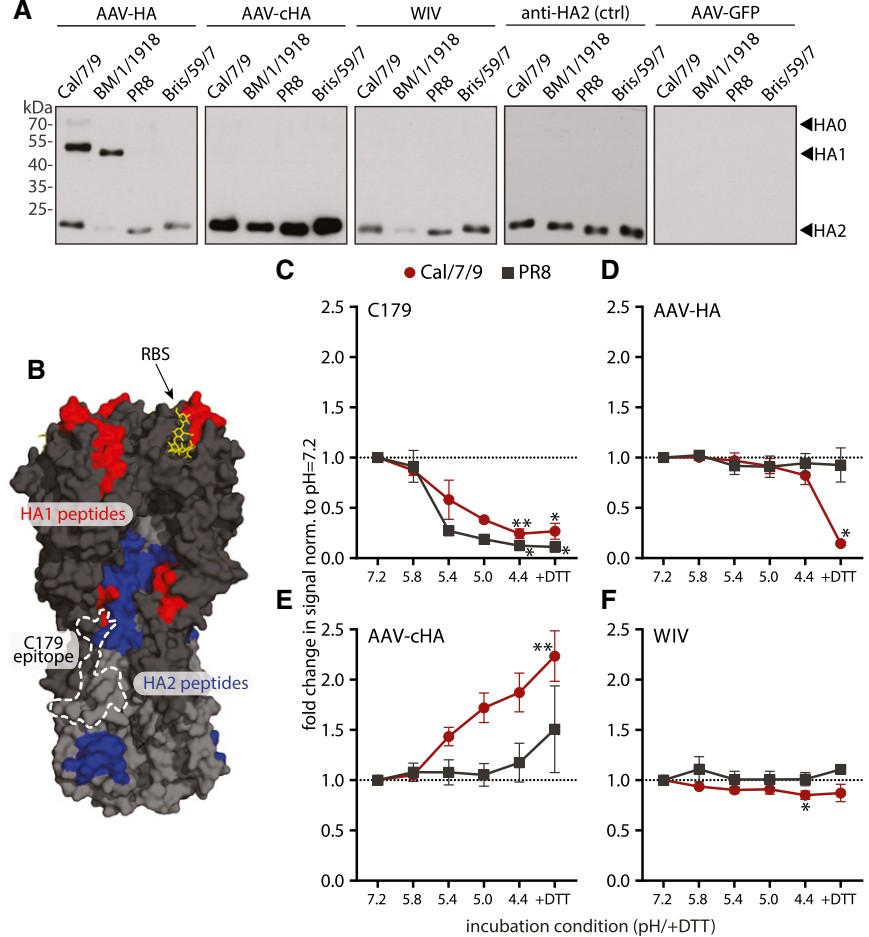

**Figure 3.  AAV-HA, AAV-cHA, and WIV induce distinct HA-specific antibody profiles.**

A    Immunoblot analysis of purified H1N1 viruses A/California/7/2009 (Cal/7/9), A/Brevig Mission/1/1918 (BM/1/1918), A/Puerto Rico/8/1934 (PR8), or A/Brisbane/59/2007 (Bris/59/7) separated under reducing and denaturing conditions. Uncleaved HA0 and cleavage products HA1 and HA2 were detected with pooled pre-challenge sera (AAV-HA, -cHA, -GFP, WIV *n* = 18, AAV-NP *n* = 11) as indicated (*n* = 3).

B    Significant peptides identified in the epitope screen with AAV-HA pooled pre-challenge serum ((AAV-HA, -cHA, -GFP, WIV *n* = 18, AAV-NP *n* = 11; data shown in Fig EV2) were mapped onto HA1 (red) and HA2 (blue) of the trimeric HA (PDB 3UBE generated with PyMol) (*n* = 3, technical duplicates). The position of the RBS (yellow) and the C179 epitope (dashed line) is marked.

C–F  Binding of C179 (C), or antibodies present in AAV-HA (D), AAV-cHA (E), or WIV (F) pooled pre-challenge sera (*n* = 18 mice per group) to Cal/7/9 or PR8 virus after incubation of the virions at pH = 7.2, 5.8, 5.4, 5.0, 4.4 or 4.4 + DTT to induce conformational changes in HA or remove the HA1 subdomain. Mean ± SD (*n* = 3, in technical triplicates). Statistical significance between pH = 7.2 and other conditions was determined using Kruskal–Wallis test with Dunn's multiple comparison testing (*\*P* < 0.05, *\*\*P* < 0.01).

Source data are available online for this figure.

recently been revised to acknowledge that all broadly reactive HA antibodies require activation of FcγR to mediate protection (DiLillo et al, 2016). Thus, we investigated whether both head and stalk antibodies contained in the AAV-HA sera could activate FcγR. To quantify the influence of the HA-head domain on FcγR activation, we assessed serum antibodies with transfected target cells expressing either complete HA or a stalk-only HA (Fig EV3C and D). In fact, the signal observed with the complete HA was much more pronounced in comparison with HA-stalk-only expressing cells, indicating that AAV-HA induces FcγR-activating stalk and head antibodies (Fig 4C). As expected, as AAV-cHA-induced sera contained mainly HA-stalk antibodies (Fig 3A), signal intensities were comparable with both antigens, whereas WIV antiserum stimulated very poorly (Fig 4C). These results indicate that AAV-HA, AAV-NP, and AAV-cHA vaccines potently induce broadly reactive FcγR-activating antibodies. Furthermore, AAV-HA, in contrast to AAV-cHA, induced FcγR-activating antibodies against the HA-stalk and HA-head regions.

### AAV-HA, AAV-cHA, and AAV-NP protect mice from homologous and heterologous challenge

To assess homologous and heterologous protection, groups of mice were challenged with lethal doses of the divergent H1N1 viruses Cal/7/9 or PR8 that differ in their HA1 amino acid sequence by more than 25% (Fig EV2A). After Cal/7/9 challenge, AAV-HA- and AAV-NP-immunized animals were completely protected, while all animals of the negative control group (AAV-GFP) succumbed to the infection (Fig 5A). AAV-cHA and WIV protected four out of five animals. Only AAV-HA-immunized mice did not show any sign of weight loss, likely corresponding to the presence of neutralizing antibodies (Figs 5B and EV4A, and Table 1). All other groups showed about 10% mean maximum weight loss. Nevertheless, surviving animals recovered quickly and were able to clear the virus at the end of the 14-day monitoring period (Figs 5B and EV4B).

To evaluate protection against a heterologous H1N1 strain, mice were challenged with PR8. Although lethal for control animals (AAV-GFP), the dose used for the first challenge experiment did not reveal differences in survival rates of vaccinated groups (Fig 5C). However, AAV-HA-, AAV-cHA-, and AAV-NP-immunized animals showed only moderate mean maximum weight loss (3–4.5%), which was significantly less than in the AAV-GFP-vaccinated animals (21%) (Figs 5D and EV4C). In contrast, WIV-immunized animals experienced up to 10% mean maximum weight loss. As with Cal/7/9 challenge, those animals were finally able to clear the virus, despite the absence of pre-existing neutralizing antibodies against PR8 (Fig EV4D, Table 1). This result indicated that the AAV-vectored vaccines were superior to the inactivated vaccine against heterologous challenge in reducing the extent of weight loss. To evaluate the effect of the HA-based AAV vector vaccines on protection, we conducted another PR8 challenge with a higher inoculation dose. Here, the majority of mice immunized with AAV-HA or AAV-cHA were protected from severe disease and death (71%; 5/7), which was also reflected by mean maximum weight loss of 10% (Figs 5E and F, and EV4E). In contrast, WIV-immunized and negative control (AAV-GFP) animals were not protected (Fig 5E and F). Interestingly, at 3 days post-infection with PR8, virus titers in the lungs were comparable among all groups. At the end of the

monitoring period, however, only AAV-HA and AAV-cHA vaccination led to a significant reduction in mean lung virus titers compared to the other groups (Fig EV4F). In summary, these results indicate that three doses of AAV-HA or AAV-cHA were superior to two doses of WIV in reducing mortality and disease severity against heterologous influenza challenge in mice.

### AAV-vectored vaccines reduce disease severity in ferrets

The protective efficacy of AAV-vectored vaccines was finally assessed in ferrets, as they most closely reproduce the course of human influenza (Enkirch & von Messling, 2015). While focusing on HA-based vaccines, groups of four animals were vaccinated three times in 4-week intervals with AAV-HA, AAV-cHA, or AAV-GFP (Fig 6A). As control, one group was given two doses of the commercial quadrivalent influenza vaccine (QIV) of season 2017/18 by the intramuscular route. The H1N1 component of QIV of this season contained A/Michigan/45/2015 (H1N1)pdm which is closely related to Cal/7/9. In fact, ferret convalescent sera raised against Cal/7/9 react undistinguishably with A/Michigan/45/2015 (WHO, 2018c). All animals except for the AAV-GFP group developed influenza virus-specific antibodies (Fig 6B). However, a statistically significant continuous increase of antibody titers over time was only seen in the AAV-HA group. As observed in mice, HAI$^+$ and MN$^+$ antibodies against the homologous Cal/7/9 viruses were only found in AAV-HA-immunized animals (Fig 6B). To exclude that QIV immunization induced HAI$^+$ and MN$^+$ antibodies specific for A/Michigan/45/2015 virus but not Cal/7/09, we repeated these assays with the respective virus. As with Cal/7/9, only AAV-HA-immunized ferrets mounted MN$^+$ or HAI$^+$ antibodies against A/Michigan/45/2015 (Fig EV5A and B).

Ferrets were sublethally challenged with $10^5$ TCID$_{50}$/animal of the early (H1N1)pdm isolate A/Mexico/InDRE4487/2009, which can cause severe disease in ferrets (Meunier et al, 2012). During the 3-day challenge period, clinical signs of influenza were monitored, and weight and temperature were measured. At day 3 post-infection, animals were sacrificed, and nasal turbinates, trachea, and lungs were collected and processed for virus titration and/or histopathological examination (Fig 6A). All animals developed classical signs of influenza disease starting on day 1, including serous nasal exudate, congestion, frequent sneezing, wheezing, and depression (Fig 6C). However, ferrets receiving AAV-HA or AAV-cHA recovered more quickly as evidenced by a return to normal activity levels and improvement of respiratory signs compared to QIV and AAV-GFP-immunized groups. Moreover, for AAV-HA-immunized animals, a milder clinical course was associated with high MN$^+$ and HAI$^+$ antibody titers (Fig EV5D). Upon infection, body temperature increased in all animals, which continued to increase in QIV-immunized animals until day 3 post-infection, while only a short fever peak was seen in the AAV vector vaccinated groups (Fig 6D). Moreover, maximum weight loss of QIV-immunized animals was more severe (up to 10%) compared to the other groups (3–6%; Fig 6E).

Virus was isolated from the respiratory tract of all animals to various extents, indicating that no sterile immunity was induced (Fig 6F). Significantly, AAV-HA-immunized ferrets had reduced virus titers in tissue homogenates of nasal turbinates, trachea, and lung, while all other groups showed higher virus titers (Fig 6F).

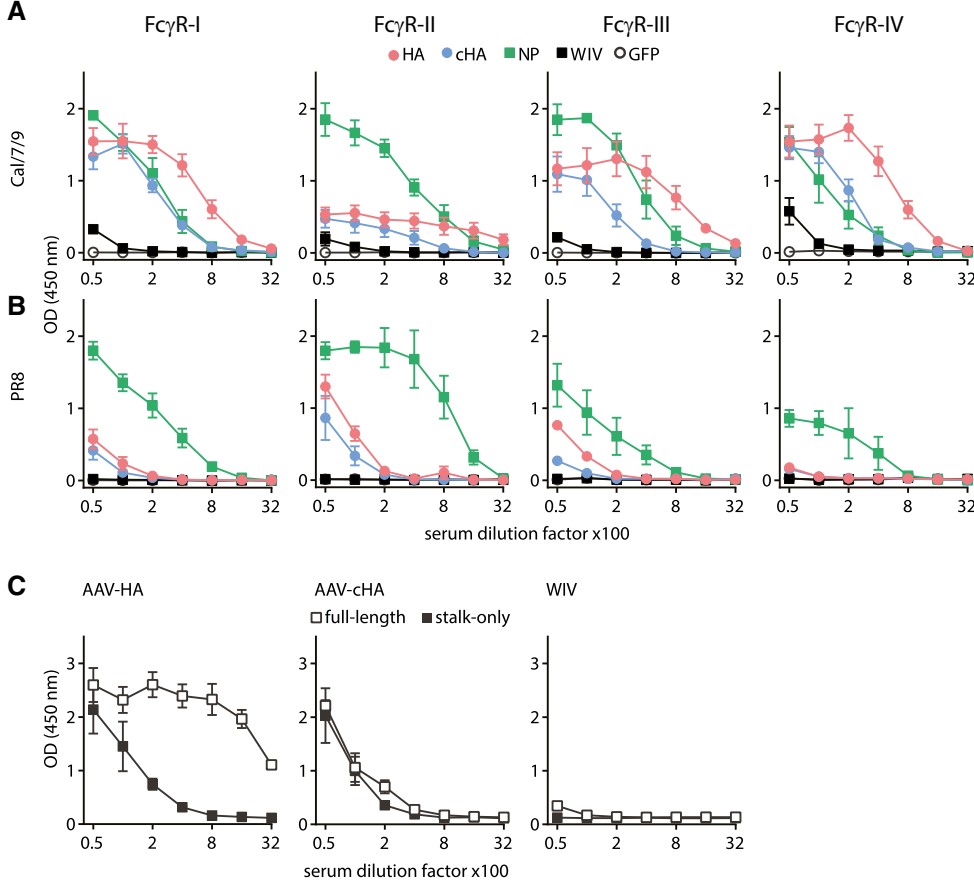

**Figure 4. AAV-vectored vaccines but not WIV induce broadly reactive FcγR-activating antibodies.**

A, B  Influenza virus-specific activation of the murine FcγRI, FcγRII, FcγRIII, and FcγRIV by antibodies in pooled pre-challenge sera (AAV-HA, -cHA, -GFP, WIV *n* = 18, AAV-NP *n* = 11 mice per group). MDCKII cells were infected with Cal/7/9 (A) or PR8 (B) before mouse serum and FcγR-expressing reporter cells were added. Influenza-specific Fc-FcγR interaction induces the production of IL-2 in the FcγR-expressing reporter cell, which was measured by anti-IL2 ELISA (*n* = 3, technical duplicates). Mean ± SE.

C  HA-specific activation of FcγR by AAV-HA, AAV-cHA, or WIV pooled pre-challenge sera (*n* = 18 mice per group). MDCKII cells were transfected with wild-type HA (pAAV-HA) or a stalk-only HA (pAAV-mHL1) before mouse serum and FcγR-I-expressing reporter cells were added (*n* = 2, technical duplicates). Mean ± SE.

Furthermore, neutralizing antibody titers in AAV-HA-immunized animals correlated with reduced viral titers in the nasal turbinates (Fig EV5E). Immunohistochemical staining detected influenza virus antigen in the lungs of all animals yet with clear differences between groups. The strongest differences were observed between the AAV-HA-immunized ferrets, with only rare virus antigen in the submucosal glands and bronchial epithelial cells when compared to the AAV-cHA- or AAV-GFP-immunized ferrets, which had massive amounts of antigen-expressing cells in both locations (Figs 6G and H, and EV5F and G). The load of virus antigen in QIV-vaccinated ferrets was in between the AAV-HA-vaccinated animals and the two other groups (Fig 6G and H). Also, all ferrets developed signs of an influenza virus-induced bronchio-interstitial pneumonia, although at varying degrees. Damage of bronchi and bronchioles was most pronounced in AAV-GFP-immunized animals (Fig EV5H), while lesions of lung interstitium as well as submucosal glands were more severe in AAV-cHA- and QIV-immunized animals (Figs 6I and J, and EV5I). Ferrets receiving AAV-HA had less severe lesions in the conducting airways and lung compared to animals from the other

groups (Fig EV5H–J). These results indicate that AAV-HA and AAV-cHA immunization reduced the disease severity after homologous influenza virus infection in ferrets. However, only with AAV-HA this was also associated with reduced virus replication and pathogenesis.

# Discussion

Conventional inactivated vaccines fail to elicit broadly protective antibodies recognizing the HA stalk, as the dominant immune response in mice (Altman *et al*, 2015) and humans (Ellebedy *et al*, 2014) is directed against variable antigenic sites in the HA head. Development of strategies which shift the focus of the immune response toward conserved HA epitopes has thus become an important research objective (Krammer, 2017). Toward this goal, sequential immunizations with different chimeric HA or headless HA were shown to trigger broadly reactive stalk antibodies (Impagliazzo *et al*, 2015; Wohlbold *et al*, 2015; Yassine *et al*, 2015; Nachbagauer

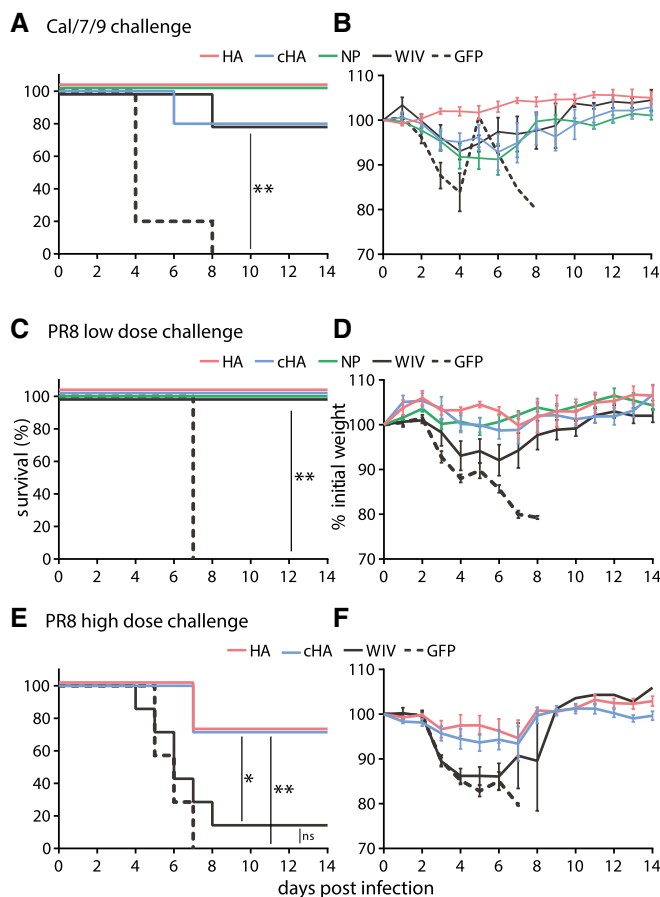

**Figure 5. AAV-HA, AAV-NP, and AAV-cHA induce broad protection against H1N1 challenge.**

A–F Mice were vaccinated three times with AAV vectors or two times with WIV and then challenged with a lethal dose of homologous influenza virus (Cal/7/9) (A, B; $n = 5$) or two different lethal doses of the heterologous PR8 strain (C, D; $n = 6$ and E, F; $n = 7$). Survival was monitored over a 14-day period and is depicted as Kaplan–Meier plot (A, C, E). Weight loss was determined during the 14-day period, which is shown in relation to the weight at day 0 (B, D, F). Mean $\pm$ SE. Statistical significance between negative control group (AAV-GFP) or WIV group (E), and each vaccine group was determined using log-rank (Mantel–Cox) test (*$P < 0.05$, **$P < 0.01$).

system for use in human gene therapy and were successfully used in preclinical active immunization studies against several infectious diseases (Nieto & Salvetti, 2014). Furthermore, AAV vectors can be produced in cell culture, stored at room temperature, and applied without needles into the nose, offering possible solutions to challenges regarding production and application of the current vaccine (Grimm & Buning, 2017).

Using the AAV vector, our study confirmed that sequential immunization with different cHA focuses the humoral response to the HA stalk and provides heterologous protection. However, it was unexpected to find that also AAV-vectored full-length HA was able to elicit not only neutralizing, but also substantial levels of HA-stalk-specific antibodies. This conclusion is supported by the observations that AAV-HA-induced antibodies (i) detected the HA2 domain in the immunoblot assay (Fig 3A), (ii) bound to peptides derived from the HA-stalk region (Fig 3B), and (iii) reacted with a natively folded HA-stalk construct, as measured by in-cell ELISA (Fig EV2C). Hence, AAV-HA immunization induced conformational and non-conformational HA-stalk antibodies. Additionally, AAV-HA immunization had in contrast to AAV-cHA the benefit of mounting HAI$^+$ and MN$^+$ antibodies toward the HA head of the homologous H1N1 virus. The importance of broadly reactive antibodies against the HA head employing novel modes of action has been demonstrated recently (Bangaru *et al*, 2019; Watanabe *et al*, 2019). Such antibodies might be induced by AAV-HA as well. AAV-HA and AAV-cHA also induced Fc$\gamma$R-activating antibodies targeting homologous and heterologous viruses (Fig 4), which is a desirable property as most broadly reactive antibodies require activation of FcR to mediate protection (DiLillo *et al*, 2014, 2016). Interestingly, AAV-HA immunization triggered Fc$\gamma$R-activating antibodies not only toward the HA stalk, but also against epitopes present only in the context of full-length HA, likely located in the HA-head domain (Fig 4). Consequently, we observed that immunization with AAV-HA conferred sterile immunity against a homologous virus, and heterologous protection toward a drifted H1N1 virus. The precise extent of Fc$\gamma$R-mediated effector functions in the observed heterologous protection by AAV-HA and AAV-cHA remains to be determined. While Cal/7/9-specific Fc$\gamma$R-activating responses were more potent with the activating Fc$\gamma$R (I, III, and IV), which are associated with induction of ADCC, highest titers of heterologous Fc$\gamma$R-activating antibodies were reached with the inhibitory Fc$\gamma$R-IIB. Thus, a regulatory role of these antibodies might be indicated, favoring for example enhanced B-cell responses (Wang *et al*, 2015).

The induction of broadly binding but non-neutralizing serum antibodies by intranasal WIV vaccination in our study recapitulates and extends observations by others (Bhide *et al*, 2019). It was actually suggested that protection mediated by intranasally applied WIV involves cellular and mucosal immunity (Dong *et al*, 2018; Bhide *et al*, 2019) including elevated levels of IgA that might have conferred protection against homologous and mild heterologous challenges in our study. However, WIV-induced responses did not protect against a harsh heterologous challenge, which correlated with the failure to induce heterologous Fc$\gamma$R-activating antibodies.

Previous studies described a restriction of the immune response to HA within the same antigenic group of influenza A viruses (Krammer *et al*, 2013; Margine *et al*, 2013), which was also apparent with the AAV-HA or AAV-cHA immunization regimens tested so far (Fig 2). Hence, we suggest evaluating in future work

*et al*, 2017). Other challenges associated with the vaccine manufacturing cycle including egg-based production might, however, not be overcome by these candidate antigens (Impagliazzo *et al*, 2015; Yassine *et al*, 2015, Register ECT, 2017). Thus, it has been appreciated that innovative delivery platforms such as viral vectors could be an important element on the quest for a universal and long-lasting influenza vaccine (Erbelding *et al*, 2018; Ortiz *et al*, 2018). Commonly used viral vectors have proven their efficacy as influenza vaccine carriers in multiple studies (de Vries & Rimmelzwaan, 2016). However, most "traditional" vectors express viral gene products and/or induce strong inflammatory responses, causing safety concerns. Reports regarding genotoxicity have been limited to wild-type AAV infection (Nault *et al*, 2015), while AAV vectors show an advantageous safety profile (Buning & Schmidt, 2015). AAV vectors represent the first EMA- and FDA-approved (FDA, 2018) vector

AAV-vectored combinations of HA components, e.g., H1 and H3, or inter-group stalk consensus constructs for the capacity to expand protection to both antigenic groups. Our findings suggest that AAV vectors might also be explored as useful carriers for other conserved viral antigens such as NA or M2e (Krammer et al, 2018).

Unexpectedly, none of three tested headless HA appeared to be immunogenic or conferred protection when expressed from an AAV vector (Fig EV5G–J). Other headless HA-based approaches relied on heterologous prime/boost regimens (DNA/VLP, i.n./i.m.), carrier particles, or adjuvants to increase immunogenicity and stimulate

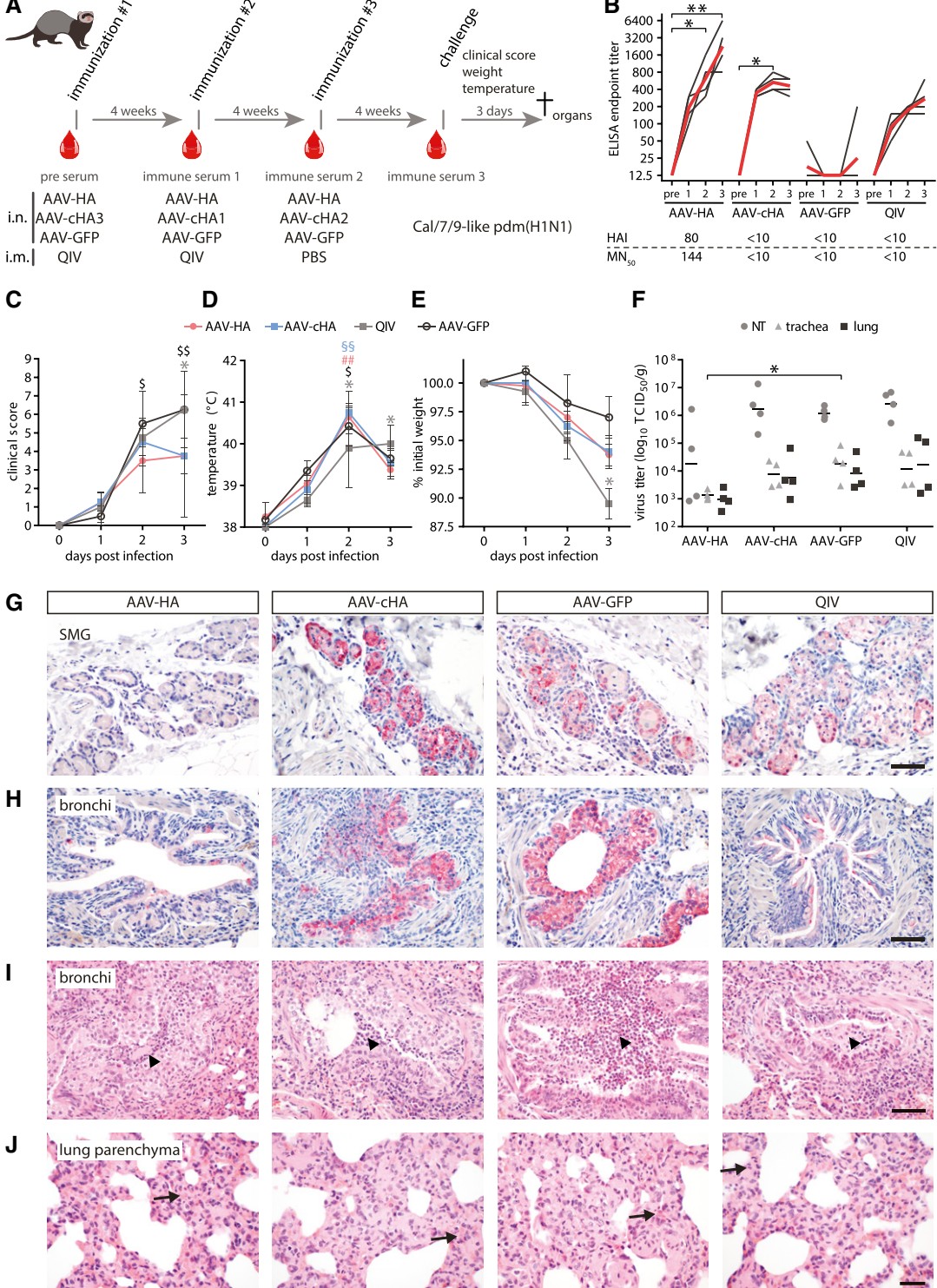

Figure 6.

◄

immune responses (Steel *et al*, 2010; Impagliazzo *et al*, 2015; Yassine *et al*, 2015). A recent study suggested that poor immunogenicity of headless HA was associated with the inability to robustly induce T-follicular helper cells, thus limiting B-cell responses, which could only be overcome by linkage to an immunogenic carrier (Tan *et al*, 2018). Interestingly, lack of immunogenicity was also found with a MVA vector expressing headless HA (Hessel *et al*, 2011). This could be circumvented by vectored co-expression of the strongly immunogenic NP, probably allowing for more efficient induction of HA-stalk antibody-secreting cells after intermolecular help by NP-specific CD4 T cells (Hessel *et al*, 2011; Alam *et al*, 2014). In our study, a bivalent AAV-mHL1 + AAV-NP vaccine had no such effect, probably because mHL1 and NP had to be expressed from two AAV vectors due to the restricted packaging capacity, thereby limiting simultaneous uptake of both antigens into one cell. Thus, an AAV-vectored approach solely based on headless HA does currently not seem to be feasible.

AAV-NP immunization induced a remarkably broad antibody response covering all tested influenza viruses and capable to activate FcγR, despite the fact that NP is considered a major stimulus of cytotoxic T-cell immunity (Sipo *et al*, 2011). Previous studies suggested a protective capacity of NP-specific FcγR-activating antibodies (Vanderven *et al*, 2016). However, the exact mechanism remains unclear, since these antibodies target a viral antigen supposedly localized in the interior of the virion or host cell (Bodewes *et al*, 2013). Besides ADCC induction, FcγR regulates innate and adaptive immunity, including antibody responses themselves by regulation of B-cell selection, maturation, and survival (Bournazos & Ravetch, 2017). The finding that AAV-NP induced antibodies potently activated the murine inhibitory FcγRIIB might suggest rather a regulatory role of these antibodies. However, both humoral and cellular immunity will likely contribute to NP-mediated protection. Further experiments are needed to extend knowledge on involved protective mechanisms (Carragher *et al*, 2008; Mullarkey *et al*, 2016; He *et al*, 2017).

In ferrets that are considered the animal model recapitulating most aspects of human influenza, we could for the first time show that active immunization with AAV-HA or AAV-cHA confers protection against a homologous viral challenge. However, only AAV-HA-immunized ferrets showed reduced virus replication and lung pathology, likely due to induction of neutralizing antibodies against (H1N1)pdm viruses in these animals. Though AAV-cHA immunization symptomatically protected ferrets as indicated by a reduced clinical score, no reduction in virus titer was seen. This is in contrast to a study where vaccination with different cHA expressed by influenza B-, VSV-, or AdV vectors reduced viral titers (Krammer *et al*, 2014). Unexpectedly, AAV-GFP-immunized ferrets, though showing an increased clinical score, lost least weight during challenge. It can only be speculated on the reasons, but this might be a consequence of the lack of protective antibody allowing for more efficient replication and severe damage of the upper respiratory tract compared to the other groups, which might have resulted in a different manifestation of disease. This topic clearly requires further investigations.

The observed lack of protection in QIV-immunized ferrets is counterintuitive. However, this finding recapitulated observations by others and likely depended on a failure to induce HAI$^+$ antibodies by the commercial, non-adjuvanted human vaccine (Baras *et al*, 2011; Nachbagauer *et al*, 2017; Liu *et al*, 2019). Neither in ferrets nor mice, change of the AAV vector capsid was required to achieve high influenza-specific antibody titers upon intranasal vaccination, even in the presence of AAV9-neutralizing serum antibodies. This finding is in line with observations by others (Limberis & Wilson, 2006). Seroprevalence of AAV9-neutralizing antibodies in humans is generally low (Boutin *et al*, 2010), suggesting an advantage over other viral vectors in the context of pre-existing immunity which critically influences transduction efficiency (Nachbagauer *et al*, 2015; Ryder *et al*, 2015). Current production capacities for AAV vectors allow the production of batches suitable for midscale clinical studies, but recent advances give a perspective that future large-scale AAV vector production and purification platforms will be able to match the demand for higher number of AAV vector-based vaccine doses (Nass *et al*, 2017).

Here, we show that vaccine delivery by the AAV-HA and cHA vectors resulted in the favorable induction of broadly reactive and FcγR-activating antibodies, which was not achieved with inactivated vaccine given either by the intranasal (this study) or intramuscular routes (Angeletti *et al*, 2019). Important factors leading to this finding included most likely continuous generation of antigen by respiratory cells after mucosal immunization, and the lung-specific environment, which influences processing and presentation of antigen to cells of the adaptive immune system (Angeletti *et al*, 2017). We speculate that this vectored delivery resulted in more effective priming of rare B cells recognizing conserved epitopes in the HA stalk (Ellebedy *et al*, 2014). Hence, AAV vectors might be usable in a naïve human population to efficiently prime a broadly reactive B-cell memory, or to boost and maintain such a response in influenza experienced individuals (Andrews *et al*, 2015). In conclusion, the

data suggest a large potential for the development of AAV vectors into a carrier for a broadly protective influenza vaccine.

## Materials and Methods

### AAV vector design and production

All antigens were based on HA or NP of Cal/7/9 (H1N1)pdm virus and were codon-optimized to mammalian gene expression and *de novo* synthesized (GeneArt®, Thermo Fisher Scientific, Regensburg, Germany). The construction of AAV-HA and AV-NP has been described before (Sipo *et al*, 2011). The initial headless HA (HL) gene was constructed by replacement of the sequence between codons of cysteine52 and cysteine277 ("head") of HA1 with a tetra-glycine linker as described before (Steel *et al*, 2010). Three different chimeric HAs (cHA) were constructed by insertion of the head regions, i.e., regions cysteine$_{52}$–cysteine$_{277}$, derived from of avian influenza A viruses of the subtypes H2, H10, or H13 into the respective region of the headless HA, thereby removing the glycine linker. Modified headless proteins 1 and 2 (mHL1 and mHL2) were constructed by transferring the mutations described (Impagliazzo *et al*, 2015; Yassine *et al*, 2015) into the Cal/7/9 HA. All constructs were inserted into the AAV vector plasmid in between the AAV2 ITRs, downstream of a CMV promoter and an irrelevant intron and upstream of a V5-tag and polyadenylation/termination signal. AAV9-vector stocks were prepared by triple transfection of the AAV-vector plasmid (pAAV), p5E18-VD2/9 (AAV2 Rep, AAV9 Cap), and pHelper (adenovirus helper genes; Addgene) into 293T cells for 72 h. Cells were lysed with sodium deoxycholate, and non-encapsidated DNA was digested with benzonase (Merck, Darmstadt, Germany). AAV vector particles were purified via isopycnic centrifugation through an iodixanol (Axis-Shield, Dundee, UK) step gradient and concentrated with Amicon Ultra-15 filter units (Merck). High quantity and quality of AAV9-vector stocks were verified by infectious titer determination, electron microscopy, ELISA, and TaqMan qPCR (forward primer: 5′-TGGAGTTCCGCGTTACAT AACTTAC-3′; reverse primer: 5′-CTATTGGCGTTACTATGGGAAC ATAC-3′, probe: 5′-FAM-CCTGGCTGACCGCCCAACGAC-BBQ-3′) using Platinum-Taq DNA Polymerase (Thermo Fischer Scientific, Hennigsdorf, Germany) with an initial denaturation step at 95°C—10 min followed by 40 cycles of 95°C—15 s; 60°C—20 s; 72°C—10 s in a Roche LightCycler 480. Tenfold dilutions of *Sma*I-digested pAAV plasmid DNA (1 ng/µl) were used as standard to calculate absolute amounts of AAV vector genomes per microliter.

### Cell lines and influenza viruses

293T were maintained in DMEM, and MDCKII cells in MEM supplemented with 10% FBS, L-glutamine, and penicillin/streptomycin. Cells were taken from the tissue culture collection of RKI's unit 17 and tested negative for mycoplasma contamination. Infections with influenza viruses were performed in cell culture medium with 0.2% BSA and TPCK-treated trypsin. Influenza A viruses (BSL2: A/California/7/2009 ((H1N1)pdm); A/Puerto Rico/8/1934 (H1N1); A/Widgeon/DK/66174/2004 (H2N3); X31 (H3N2); A/Panama/2007/1999 (H3N2); A/Mallard/NVP/9417/2004 (H10N7); A/Gull/MD/704/1977 (H13N6); BSL3: HA/NA-1918 × WSN/1933 reassortant

(H1N1); A/Viet Nam/1203/2004 (H5N1); A/Anhui/1/2013 (H7N9)) were propagated in 11-day-old embryonated chicken eggs. Allantoic fluid harvested from infected eggs was purified by ultracentrifugation through a 20% sucrose cushion and resuspended in PBS before protein content was determined via BCA (Thermo Fischer Scientific, Hennigsdorf, Germany). Infectious titers were determined by plaque assay on MDCKII cells. Whole-inactivated Cal/7/9 reassortant virus (X-181) (WIV) used for vaccination of mice was kindly provided by Othmar Engelhardt (National Institute for Biological Standards and Control (NIBSC), Potters Bar, UK). Briefly, virus was grown on 11-day-old embryonated chicken eggs, inactivated with β-propiolactone, and purified by ultracentrifugation through a 20% sucrose cushion. Protein content was determined by Lowry assay. A sample was deglycosylated and separated on reducing SDS–PAGE, and viral protein content was determined with Coomassie staining. Antigenicity of the preparation was verified using single-radial diffusion assay with Cal/7/9-specific sheep serum.

### Indirect immunofluorescence microscopy and flow cytometry

293T cells were transfected with AAV vector plasmids for 24 h using Lipofectamine 2000 (Thermo Fisher Scientific) according to the manufacturer's instruction. Cells were fixed, permeabilized, blocked, and immunostained with anti-V5-tag (1:250; SV5-Pk1; Bio-Rad, München, Germany), C179 (1:250; M145; TaKaRa, Kusatsu, Japan) antibody, rabbit hyperimmune serum against the influenza subtypes H2, H10, and H13 (1:500; 11688-RP01-100, 11693-RP01-100, 11721-RP01-100 Sino Biological Inc., Eching, Germany), or animal serum followed by suitable secondary antibody coupled to AlexaFluor488 (1:1,000; Thermo Fisher Scientific). Nuclei were stained with DAPI. Cells were analyzed using a LSM 780 confocal laser scanning microscope (Zeiss, Jena, Germany). For quantification of V5-tag or C179 or H2-, H10- and H13 HA-positive cells, samples prepared the same way were analyzed in a FACSCalibur cytometer (BD Biosciences, Heidelberg, Germany).

### Immunoblotting

293T cells were transfected with AAV vector plasmids for 48 h using Lipofectamine 2000 (Thermo Fisher Scientific) according to the manufacturer's instruction. Alternatively, 293T cells were transduced with AAV vector preparations at an MOI of $10^6$ by adding the amount of virus diluted in PBS directly to the cell culture medium for 72 h. Cell lysates were prepared in RIPA buffer (10 mM Tris/HCl (pH = 8), 150 mM NaCl, 0.5 mM EDTA (pH = 8), 0.1% SDS, 1% Triton X-100, protease inhibitor) with beta-mercaptoethanol and boiled, before samples were separated by reducing SDS–PAGE under denaturing conditions and transferred onto nitrocellulose membrane. Western blotting was performed by incubation with anti-V5-tag (1:1,000; SV5-Pk1) antibody and suited secondary antibody coupled to horseradish peroxidase (HRP) (1:10,000; Agilent Technologies, Santa Clara, USA). Equal loading of samples was controlled with GAPDH antibody (1:1,000) staining.

4–8 µg per lane of sucrose-purified influenza viruses was separated on SDS–PAGE under denaturing conditions and transferred onto nitrocellulose membrane. Amount loaded per lane was

normalized to yield equal intensities of HA2 with an anti-HA2 antibody (1:500; orb10765; Biorbyt, Eching, Germany). Western blotting was performed by incubation with mouse serum (1:500) and suited secondary antibody coupled to HRP (1:10,000; Agilent Technologies). Blots were developed on X-ray films (Thermo Fischer Scientific) after addition of SuperSignal™ West Dura Extended Duration Substrate (Thermo Fischer Scientific).

## Mouse serum and lung homogenate ELISA

For serum ELISA, MaxiSorb™ ELISA plates (Thermo Fischer Scientific) were coated with 4–8 µg/ml purified influenza virus or approximately $1 \times 10^{10}$ vg/ml AAV9 empty capsids in 50 µl 50 mM carbonate/bicarbonate buffer (pH = 9.6) overnight at 4°C. Plates were blocked with 3% skim milk in $PBST^{0.05\%}$ before twofold serum dilutions in 50 µl blocking buffer were added and plates were incubated 90 min at 37°C. For lung homogenate ELISA, plates were coated as mentioned above. Plates were blocked with 1% BSA in $PBST^{0.05\%}$, before 50 µl of a twofold dilution of lung homogenates in $PBST^{0.05\%}$ was added for 1 h at 37°C. After washing, plates were incubated with anti-mouse-IgG-HRP (1:1,000; Agilent Technologies) or anti-mouse-IgA-HRP (1:1,000; Thermo Fischer Scientific) antibody for 45 min at 37°C before plates were washed again and 1-Step™ Ultra TMB-ELISA Substrate Solution (Thermo Fischer Scientific) was added for 1–10 min until color reaction was stopped with 1 M $H_2SO_4$. Optical density was measured at 450 nm ($OD_{450\ nm}$), and the area under the curve was determined using GraphPad Prism software.

## HA conformational change ELISA

MaxiSorb™ ELISA plates (Thermo Fischer Scientific) were coated with 4–8 µg/ml purified influenza virus in 50 µl carbonate/bicarbonate buffer (pH = 9.6) overnight at 4°C. 100 µl per well of acetate buffer at different pH (7.2, 5.8, 5.4, 5.0, 4.4, and 4.4 + 0.1 M DTT) was added at RT for 30 min. Plates were washed two times, and ELISA was performed as described above.

## In-cell ELISA

MDCKII cells were transfected with pAAV-HA, pAAV-mHL1 + transmembrane region or with pcHA3 for 48 h using Lipofectamine 2000 (Thermo Fisher Scientific). Cells were fixed, permeabilized, blocked, and incubated with AAV-HA, AAV-cHA, or AAV-GFP mouse serum (1:250), or C179 (1:250) for 1 h at 37°C. After washing, cells were incubated with anti-mouse-IgG-HRP (1:1,000; Agilent Technologies) for 1 h at 37°C before plates were washed again, and 1-Step™ Ultra TMB-ELISA Substrate Solution (Thermo Fischer Scientific) was added for 10 min until color reaction was stopped with 1 M $H_2SO_4$. Optical density was measured at 450 nm ($OD_{450\ nm}$), and values were normalized to AAV-GFP-negative control values.

## Immunoperoxidase monolayer assay

Immunoperoxidase monolayer assay (IPMA) was used to determine total antibody titers in ferret sera against Cal/7/9 virus and was done as described previously (Walz *et al*, 2018). Briefly, MDCKII cells were infected with Cal/7/9 at a multiplicity of infection of 0.01

and incubated at 37°C for 2 days. Cells were then washed with 30% PBS in $H_2O$, air-dried, and heat-fixed at 65°C for 8 h. Twofold serial dilutions of duplicate serum samples were added to the plate, followed by incubation with a peroxidase-labeled anti-ferret IgG antiserum (1:750; Bethyl Laboratories Inc., Montgomery, USA). Bound antibody was visualized using amino-ethyl-carbazole, and antibody titers were expressed as the reciprocal value of the last dilution with positive staining.

## Microneutralization assay

Influenza MN assay was done in essence as described (He *et al*, 2015) with the exception that $3.5 \times 10^4$ PFU of influenza virus was used per 96 well to examine mouse sera and $2 \times 10^2$ PFU per 96 well for ferret sera. Detection of influenza antigen was done with anti-influenza A polyclonal antibody (1:1,000; 5315-0064; Bio-Rad) and suited secondary antibody coupled to HRP (1:1,000; Agilent Technologies). AAV MN was done as described (Meliani *et al*, 2015) with the exception that AAV-GFP was used as reporter virus and GFP-positive cells were quantified by flow cytometry in a FACSCalibur cytometer (BD Biosciences). Percent inhibition was calculated at each dilution step, and the $MN_{50}$ (dilution step at which 50% inhibition was measured) was determined using nonlinear fit of the inhibition curves [(Inhibitor) vs. response-variable slope (four parameters)] in the GraphPad Prism software. The limits of detection were dilutions of 1:100 (for AAV MN), 1:40 (for mouse influenza MN), and 1:10 (for ferret influenza MN), respectively.

## Hemagglutinin inhibition (HAI)

Animal sera were inactivated with trypsin-heat-$KIO_4$ treatment (WHO, 2002). Twofold dilutions of sera in PBS were prepared in 25 µl per 96-well. Four HAU per 25 µl PBS of influenza virus were added to each well, and samples were incubated at 37°C for 1 h. 50 µl of 1% chicken red blood cells in PBS was added to each well, and samples were incubated at 4°C for 20 min before agglutination inhibition titers were determined as lowest serum dilution with a clear agglutination clot. The limit of detection was a dilution of 1:40.

## Epitope screen

Unmodified 15-mer peptides overlapping by five amino acids (peptides & elephants GmbH, Hennigsdorf, Germany) were reconstituted in 50% DMSO in $H_2O$. MaxiSorb ELISA plates (Thermo Fischer Scientific) were coated with 1 µg of peptide in 50 µl carbonate/bicarbonate buffer (pH = 9.6) per well overnight at 4°C. Plates were blocked 1 h at 37°C with 1% BSA in $PBST^{1\%}$ before mouse serum (1:200) in 50 µl blocking buffer was added and plates were incubated at 37°C for 1 h. After washing, plates were incubated with anti-mouse-IgG-HRP (1:500; Agilent Technologies) for 45 min at 37°C, before plates were washed again and -Step™ Ultra TMB-ELISA Substrate Solution (Thermo Fischer Scientific) was added for 10–20 min until color reaction was stopped with 1 M $H_2SO_4$. Optical density was measured at 450 nm ($OD_{450\ nm}$). Values were normalized to negative control serum (AAV-GFP).

### The paper explained

#### Problem

Current seasonal influenza vaccines show low effectiveness and protection is limited to the virus strains contained within the vaccine. High morbidity and mortality caused by seasonal influenza and the risk of emergence of pandemic and/or zoonotic virus strains emphasize the urgent need for a broadly reactive vaccine.

#### Results

AAV vectors were used to deliver influenza antigens to the lung of mice and ferrets to induce protective immunity. AAV vectors expressing HA, NP, or chimeric HA were shown to protect mice from challenge with divergent H1N1 virus strains. This was associated with the induction of non-neutralizing but FcγR-activating antibodies. AAV-HA was also shown to induce protective immunity in ferrets against a homologous H1N1 challenge.

#### Impact

The results of this work demonstrate that the AAV vectors are promising carriers for a broadly reactive influenza vaccine. The vectored expression of the antigen was shown to balance the immune response toward more conserved broadly reactive epitopes within the influenza virus antigens and to induce high levels of FcγR-activating antibodies. Furthermore, the licensure of AAV vector for human gene therapy could ease further clinical development of a vaccine.

## FcγR assay

To display influenza virus surface antigens on MDCKII cells, cells were seeded into 96-well plates and either infected with influenza virus ($3.5 \times 10^3$ PFU per well) or transfected with pAAV-HA or pAAV-mHL1 plasmids for 24 h using Lipofectamine 2000 (Thermo Fisher Scientific). 50 μl of twofold serial serum dilutions in R10 medium was added to each well at 37°C for 1 h. MDCKII cells were washed once with PBS, before $2 \times 10^5$ of either BW:FcγRI-ζ, BW:FcγRIIB-ζ, BW:FcγRIII-ζ, or BW:FcγRIV-ζ reporter cells were added per well in 200 μl R10 medium, and plates were incubated at 37°C overnight. These reporter cells express one of the four murine FcγR (I, IIB, III, IV), which was fused to the CD3-ζ transmembrane and signaling domain. Thus, activation of the FcγR by immune complexes triggers IL-2 production, which is quantified by ELISA (Corrales-Aguilar *et al*, 2013; Van den Hoecke *et al*, 2017). To release intracellular IL-2 cells were disrupted by the addition of Tween 20 and resuspended. Supernatant was transferred to Maxisorb™ ELISA plates (Thermo Fischer Scientific), which had been precoated with rat anti-mouse-IL2 capture antibody (1:500; 554424, BD Biosciences), and plates were left at RT for 1 h. After washing, biotinylated-IL2 detection antibody (1:500; 554426, BD Biosciences) was added and plates were incubated at RT for 1 h. After washing again, streptavidin–HRP (1:1,000; 016-030-084, Jackson ImmunoResearch, Ely, IK) was added at RT for 30 min before plates were washed and 1-Step™ Ultra TMB-ELISA Substrate Solution (Thermo Fischer Scientific) was added for 1–5 min before reaction was stopped with 1 M $H_2SO_4$. Optical density was measured at 450 nm ($OD_{450 \text{ nm}}$).

## Animal challenge studies

All mouse and ferret immunization and challenge experiments were approved by the local committees and were conducted in accordance with national guidelines for the care and use of laboratory animals (Landesamt für Gesundheit und Soziales (Berlin, Germany) approval G0255/16, Regierungspräsidium Darmstadt (Germany) approval F107/117 and IMI17 by Regierungspräsidium Tübingen).

## Handling of mice

Female 6- to 8-week-old C57BL/6 mice were purchased from Charles River Laboratories (Sulzfeld, Germany) and housed at the animal facility of the Robert Koch Institute (Berlin, Germany). The minimally required animal numbers per group were determined using the software tool G*Power 3.1.6 (University of Kiel, Germany) with data obtained in an initial immunization study aiming at a power of 80%. Mice were anesthetized by isoflurane inhalation, before 50 μl PBS containing $10^{11}$ vg of AAV vectors was added dropwise to both nostrils. Mice were immunized three times with AAV vectors in 3-week intervals. Likewise, 20 μg of Cal/7/9 (X-181) in PBS was administered intranasally two times with the first and the second immunization of the AAV vector-vaccinated animals. Retro-orbital sinus blood samples were taken from the animals before each and after the last immunization from which serum was obtained for serological assays. Three weeks after the third immunization, animals were challenged with Cal/7/9 virus ($7.94 \times 10^3$ $TCID_{50}$/mouse, 500 $MID_{50}$) or A/Puerto Rico/8/1934 virus ($2.51 \times 10^4$ (low dose) or $5.01 \times 10^4$ (high dose) $TCID_{50}$/mouse, 100 or 200 $MID_{50}$) in 50 μl PBS. During low-dose PR8 challenge, a single AAV-GFP-immunized animal, which survived without weight loss, remained influenza virus seronegative, indicating that the inoculum was not taken up properly during challenge infection. This animal was therefore excluded from the analysis. Following infection, mice presenting ≥ 20% weight loss were anesthetized and euthanized by isoflurane overdosing and lungs were harvested. Lungs of surviving mice were harvested on day 14 post-infection. Lungs were homogenized using a Fastprep-24 tissue homogenizer (MP Biomedicals, Santa Ana, USA) equipped with matrix D-lysing matrix tubes for 40 s at 6 m/s. After clarification, 10-fold serial dilutions of the homogenates were prepared, and plaque assay was done on MDCKII cells.

## Ferrets

Sixteen 4-month-old male ferrets (*Mustela putorius furo*) were purchased from EuroFerret (Denmark) and housed at the animal facilities of the Paul-Ehrlich-Institute (Langen, Germany) and tested seronegative for circulating influenza viruses prior to the experiment. Ferrets were anesthetized by intramuscular injection of ketamine (100 mg/kg) and medetomidine (0.05 mg/kg), before a total of 300 μl PBS containing $7.5 \times 10^{12}$ vg of AAV vectors were added dropwise to both nostrils. The anesthesia was then reversed by subcutaneous injection of atipamezole (0.2 mg/kg). Ferrets were immunized three times with AAV vectors in 4-week intervals. One human dose per ferret of quadrivalent inactivated influenza vaccine of season 2017/18 containing 15 μg HA per vaccine component (Influsplit Tetra®, GSK, Brenford, USA) was given intramuscularly into the posterior limbs two times in parallel with the first and the second AAV vector vaccine. Blood

samples were collected before each and after the last immunization from which serum was obtained for serological analysis. Four weeks after the third immunization, animals were challenged intranasally with $1 \times 10^5$ TCID$_{50}$ of A/Mexico/InDre4487/2009 (H1N1)pdm in 200 µl Opti-MEM.

Ferrets were monitored daily for signs of disease using a 0-1-2 scale for each activity (normal, calm, depressed), respiratory signs (sneezing, nose exudate, and congestion), and general clinical signs. Zero indicates minimal deviation from the physiological state, 1 indicates moderate nasal discharge, congestion, and/or occasional sneezing and/or calm temper, while 2 indicates severe nasal discharge and/or labored breathing, dyspnea, and frequent sneezing and/or depressed manner (Walz *et al*, 2018). Final clinical scores were calculated by summation of activity, respiratory signs, and general clinical sign records per ferret per time point. Changes in body weight and body temperature were measured. On day 3 post-infection, animals were anesthetized by injection of ketamine (100 mg/kg) and medetomidine (0.05 mg/kg), exsanguinated, and nasal turbinates, trachea and lung were harvested. Nasal turbinates and samples of the trachea and the lung were homogenized, and virus titers were determined by TCID$_{50}$ assay as described before (Walz *et al*, 2018). Lung samples were immersion-fixed in formalin for histopathological examination.

### Lung histopathology and immunohistochemistry

Formalin-fixed trachea and lung samples were embedded in paraffin, cut in 2-µm sections, and stained with hematoxylin and eosin (H&E) after dewaxing in xylene and rehydration in decreasing ethanols. Lung sections were microscopically evaluated by board-certified veterinary pathologists (KD, ADG) in a blinded fashion to assess character and severity of pathologic lesions using lung-specific inflammation score parameters for quantifying influenza virus-induced pneumonia as described (Dietert *et al*, 2017). These parameters included severity of (i) interstitial pneumonia with infiltration by macrophages, lymphocytes, neutrophils (ii) bronchitis, (iii) epithelial necrosis of bronchi, alveoli, and submucosal glands, (iv) perivascular lymphocytic cuffing, and (v) hyperplasia of type II pneumocytes.

For immunohistochemical detection of influenza A virus, heat-mediated antigen retrieval was performed in 10 mM of citric acid (pH = 6.0), in which samples were microwaved at 600 W for 12 min. Lung sections were incubated with a polyclonal influenza A H1N1 antibody (1:250; 5315-0064, AbD Serotec, Puchheim, Germany) at 4°C overnight. Incubation with an irrelevant antibody at the same dilution served as negative control. Subsequently, slides were incubated with a suited secondary, phosphatase-conjugated antibody (1:500; AP-1000, Vector, Burlingame, CA, USA) for 30 min at room temperature. Afterward, triamino-tritolyl-methanechloride (Neufuchsin) was added to the slides, yielding a reddish stain upon processing by the phosphatase. Slides were counterstained with hematoxylin, dehydrated through graded ethanols, cleared in xylene, and coverslipped. Samples from non-influenza infected ferrets served as negative control for histopathological and immunohistochemical analysis. Neither one of the above-mentioned influenza infection-specific parameter nor influenza virus antigen could be detected in these control samples.

### Statistical analyses

Statistical analysis and plotting of data were performed with GraphPad Prism® Version 7.04. Statistical significance between unpaired groups was analyzed using Kruskal–Wallis test with Dunn's post-testing as indicated in the figure legends. Statistical significance between two unpaired groups was analyzed using Mann–Whitney test as indicated in the figure legends. Statistical significance between two paired groups was analyzed using Wilcoxon paired. Differences between time points were analyzed with Friedman test with Dunn's post-testing as indicated in the figure legends. Statistically significant differences between survival curves were calculated with log-rank (Mantel–Cox) test. Linear regression analysis was performed with default settings in GraphPad.

**Expanded View** for this article is available online.

### Acknowledgements

We thank Annette Dietrich and Stefanie Bessing (animal facility, RKI) for assistance with animal studies. We thank Gudrun Heins (RKI) and Yvonne Krebs (PEI) for excellent technical assistance. We are grateful to Prof. Dr. Hartmut Hengel and Katharina Erhardt (University of Freiburg, Germany) for providing the murine FcγR-cell reporter assay, as well as to Stephen Norley (Unit 18: HIV and other Retroviruses, RKI) for providing plasmids for AAV vector production as well as technical and scientific advice. We thank Lars Möller (Centre for Biological Threats and Special Pathogens, RKI) for EM pictures of AAV preparations. We like to dedicate this work to the memory of Stephen Norley, who inspired a lot of the work contained in this communication and recently tragically passed away. This project has received funding from the European Union's Seventh Framework Program for research, technological development, and demonstration under grant agreement no. 602012 (project UNISEC).

### Author contributions

DED, OP, TW, and VvM designed experiments. DED, HH, and LW performed all *in vitro* and animal experiments; KD and ADG performed the histological analysis of ferret organs; and DED and TW wrote the manuscript. All authors revised the manuscript.

### Conflict of interest

The authors declare that they have no conflict of interest.

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
