## [Review Process File · EMBO Molecular Medicine]

Adeno-associated virus-vectored influenza vaccine elicits neutralizing and Fc γ -receptor activating antibodies

Daniel E. Demminger, Lisa Walz, Kristina Dietert, Helen Hoffmann, Oliver Planz, Achim D. Gruber, Veronika von Messling, Thorsten Wolff

Review timeline:

Submission date:	27th May 2019
Editorial Decision:	27th Jun 2019
Revision received:	18th Dec 2019
Editorial Decision:	23rd Jan 2020
Revision received:	11th Feb 2020
Accepted:	12th Feb 2020

Editor: Céline Carret

Transaction Report:

1st Editorial Decision

27th Jun 2019

Thank you for the submission of your manuscript to EMBO Molecular Medicine. We have now heard back from the three referees whom we asked to evaluate your manuscript.

Overall, you will see that all referees find the study interesting. However, clarifications, explanations and additional experiments are needed to strengthen the data. Referee 1 would like to see clarifications regarding experimental analyses (n, stats and replication, blinding...) and experimental design (no AAV-NP performed in ferrets, no immune response analyses, different treatment regime, no functional assay on Ab). Referee 2 is mostly concerned about the translational limitations of the study and we would like to encourage you to discuss these. Referee 3 is concerned about the WIV part of work and wonders whether WIV was either prepared incorrectly and/or delivered incorrectly. Following cross-commenting, all referees agreed on the important nature of this work and are all supportive of revision, such as:

"A repeat of the mouse experiment is important because it will allow the authors to address concerns on the statistics/power, allow the authors to compare the levels of inflammatory mediators following challenge and to adjust the set up of the WIV control group (e.g. intramuscular with prime boost, which is the standard route of administration in human and the regimen recommended for infants)." However, we will not ask you to repeat any ferret experiments.

We would therefore welcome the submission of a revised version within three months for further consideration and would like to encourage you to address all the criticisms raised as suggested to improve conclusiveness and clarity. Please note that EMBO Molecular Medicine strongly supports a single round of revision and that, as acceptance or rejection of the manuscript will depend on another round of review, your responses should be as complete as possible.

***** Reviewer's comments *****

Referee #1 (Comments on Novelty/Model System for Author):

The statistical analysis is well describe in all figures, and graphs. However, there are an important number of figures reporting only $n=2$, and only two dots in the figure! [results without SD or SEM],, and $n=3$ showing only 3 data points. Does this mean that no technical replicates were included in the assay? It is not appropriate to report results without technical replicates. If these results are available, then authors should show ALL data sets in the figure.

It is not evident whether the mice experiments have been carried out with enough power calculation to assess the differences. Could the authors provide such analysis? Additionally, there are no indications whether the data represent independent experiments (ie different immunization days), or just one group of animals challenged with the candidate vaccines.

Referee #1 (Remarks for Author):

Demminger and colleagues aim to provide compelling in vitro and in vivo evidence for the use of AAV as platform to develop influenza vaccines. As proof of principle they generated AAV viruses expressing influenza virus HA or chimeric HA, and investigate the protection potential by defining the type of antibody responses, and they evaluate the protection conferred in two challenge models: mice and ferrets.

This is highly relevant topic, with huge translational potential. The use of AAV viruses is well accepted, even for vaccine development, and there are already published studies assessing its potential to generate influenza vaccines. Nevertheless, still there is a need to better understand the type of responses elicited, and how universal this platform could be. The manuscript is well written and the figures are elegantly presented. The experimental approach is sensible, and the appropriate controls are included. I have the following comments for the authors consideration:

1. Please note my comments referring the statistical analysis, and the number of biological and technical replicates.
2. The histology analysis do not include a more quantitative scoring. Given the comparisons of this work, this analysis should be performed in a blind way.
3. AAV-NP viruses performed very well in the mice challenge (perhaps even better than the HA variants!), therefore it is not evident why they were excluded from the ferret experiment.
4. A major limitation of this study is the lack of analysis of immune responses at the very least in the mouse model. There is no indication on the inflammatory parameters, as well as the recruitment of immune cells, and whether differences could be found between the AAV constructs.
5. Fig 6, why the treatment regime was different between the commercial vaccine and those under consideration in this study. Without an strict side-by-side comparison it is difficult to assess whether the new AAV vectors are indeed better.
6. Fig 4, although it is relevant to detect the presence of these antibodies, authors should also show whether they are functional. Assays should be performed to determine whether the antibodies elicited upregulate antiviral effect upon activation of Fc γ R.

Referee #2 (Comments on Novelty/Model System for Author):

Except for the medical impact, I consider the experimental setups of high technical quality. I do have reservation on the eventual clinical applicability of an AAV vectored influenza vaccine.

Referee #2 (Remarks for Author):

Demminger and co-authors report on the generation, immunogenicity and protective potential of a set of AAV vectors that can express influenza A hemagglutinin and derivatives thereof. Wild type, chimeric as well as stalk-only constructs based on Cal/09 HA were cloned into the AAV vector. Transduction of 293T cells at a MOI of 106 resulted in detectable expression of all constructs by Westernblot. The headless HA constructs displayed poor reactivity with C179. In mice, 3 intranasal immunizations with AAV-HA result in higher Cal/09 binding and neutralizing serum IgG antibodies than two intranasal doses of WIV. HI and MN titers in serum of immunized mice were only detected in AAV-HA and AAV-H13H1 chimeric HA. Mouse immune sera from AAV-HA, AAV-cHA and WIV could bind to denatured HA2 of H1N1 viruses that were isolated between 1918 and 2009.

AAV-HA induced both head- and stalk-specific antibodies; WIV and AAV-cHA primarily stalk-specific antibodies. An *in vitro* cell-based FcγR-activation assay allowed to conclude that AAV-HA immune serum induced the strongest FcγR activation against Cal/09 infected cells and, although much weaker, against PR8 infected cells. In line with the poor immunogenicity of WIV in this model, WIV immune serum had a very weak capacity to activate FcγRs. FcγR activating antibodies in the immune sera were directed against HA head and stalk for the cHA vaccine and primarily directed against the HA head for the AAV-HA vaccine. AAV-HA immunized mice were best protected against a homologous challenge. Interestingly, the mouse that had to be euthanized in the challenged WIV group had no remaining virus in the lungs at the endpoint. AAV-HA and -cHA protect fully against a dose of PR8 virus challenge and partially against challenge with a double dose of PR8.

The protective potential of 3 intranasal immunizations with AAV-HA and -cHA was largely recapitulated in a ferret model. Protection against homologous challenge was clearest in 3 times AAV-HA immunized ferrets, in line with the observation that only in this group neutralizing antibodies were detected.

Overall, the experiments are very well performed and quite novel. The applicability of 3 intranasal immunizations with AAV-vectored influenza HA for seasonal influenza in human is not trivial.

Major remarks:

1. The statement in line 319 is misleading. The effect on survival of HA-specific FcγR-activating antibodies was not evaluated in the experiment. The immunized mice had likely also mounted T cell responses against the recombinant vaccine antigens and those responses may have contributed to protection. A passive serum transfer experiment followed by challenge, preferentially in wt and FcγR deficient mice could reveal an effect of FcγR-activating antibodies on protection.

2. An AAV-NP group should be included in Fig. 5C to support the conclusion that AAV-NP is superior to WIV in protection against a heterologous challenge as stated in line 328. In addition the statement should explicitly mention that 3 immunizations are compared with 2 immunizations. Does the statement on the comparison with WIV hold when protection after 2 immunizations with AAV-HA is evaluated?

3. The conclusion in line 506: AAV-vector mitigates immunodominance of strain-specific epitopes in the HA head does not appear to be supported by the data. Figures 3A and 4C (left hand panels) show a predominant response against HA1 and full length HA, respectively. A comparison with a setting in mouse or ferrets in which the WIV or QIV vaccine had actually worked, would also be needed to support the statement. I.m. injection of adjuvanted WIV or QIV will typically results in clear seroconversion in mouse and ferret models. Intranasal WIV adjuvanted with polyI:C will likely also result in a better seroconversion.

4. Figure 4C: which FcγR(s) was tested with the transfected cells as target? Please clarify in the legend.

5. The authors should discuss the limitations of the production capacity and downstream processing steps that would be required to prepare millions of doses on an annual basis of a quadrivalent AAV-vectored influenza vaccine. In addition, there is a potential safety concern. An important step in the purification process of biologicals, such as recombinant monoclonal antibodies, is a virus removal step that involves ultrafiltration. It would be interesting for the reader if the authors could address these concerns in the discussion.

Other remarks:

1. Please clarify why expression of mHL1 and mHL2 is detectable in the cytoplasm (Figure 1C and D). Expression in the cell culture medium should also be documented for these 2 constructs, which lack a TM domain.

2. Line 152: at a MOI of 1 million, it is questionable that the transduction rate can be considered as high. Please modify the statement. In addition, a dose titration range (different MOIs) should be performed to convincingly show that the different HA variant vectors can give rise to comparable expression levels based on V5-tag reactivity. The transduction experiment of which a WBlot is shown in Figure 1D, should be better described: in line 574, a plasmid transfection is described, not an AAV transduction.

3. Line 1084: "Statistical ..." belongs to the legend of 2B.

4. Please add a statement on the number of repeat experiments that were performed in the legend of figure 5 (mouse immunization and challenge).

5. Line 712 and table S1: what is the correct dose that was used for the ferret immunizations?

6. Line 671: the correct reference for the FcγR activation assay is Corrales-Aguilar et al., JIM 2013.

Referee #3 (Remarks for Author):

Comments for the authors of the EMBO Molecular Medicine manuscript number EMM-2019-10938:

The authors of the EMBO Molecular Medicine manuscript "Adeno-associated virus-vectored influenza vaccine elicits neutralizing and Fcγ receptor-activating antibodies", present interesting findings in their work with adeno-associated virus (AAV) vectors for vaccination against influenza viruses. Specifically, the authors used AAV technology to express full-length influenza virus hemagglutinin (HA), as well as chimeric HA constructs that included the globular head from H2 influenza viruses, H10 influenza viruses, or H13 influenza viruses. All of these constructs contained the stalk from the pandemic H1N1 virus A/California/7/2009. The authors also expressed headless and modified headless domains of the A/California/7/2009 stalk. As a candidate for a highly conserved epitope the authors also expressed the influenza A virus nucleoprotein (NP) within the context of the AAV vector. The authors show vaccine-induced immunity against H1N1 and H5N1 influenza viruses after vaccination with the full-length HA and immunity against Group 1 viruses (H1, H2, H5, and H13 HAs) but not group 2 viruses (H7 and H3 HAs) after vaccination with the cHAs (delivered a single time each sequentially with H13 first, H2 second, and H10 third, rather than as individual groups). Immunity against the NP recognized viruses from both groups as did antibodies against the whole influenza virus (WIV) vaccine. While antibodies were detected in these various groups, antibodies that inhibited hemagglutination (hemagglutination inhibition, HAI) and neutralized virus infectivity (microneutralization, MN) were only induced against the A/California/7/2009 virus in the AAV-HA vaccine group and against the H13N6 virus in the AAV-cHA group. Further evaluation of the antibody response identified an interaction with the HA2 domain after vaccination with AAV-HA, AAV-cHA, and WIV. These antibodies also activated Fc receptors, specifically the RI, RIII, and RIV receptors that are involved in ADCC reactions. Anti-NP antibodies (induced by the AAV-NP vaccine) reacted with Fc receptors from the RI, RII, RIII, and RIV classes.

When tested in a challenge model with A/California/7/2009, the HA and NP vaccines protected 100% of mice from death, while the cHA and WIV vaccines protected about 80% of mice. When PR8 was used for challenge, the low-dose challenge (100 MID50) did not induce any death while a high-dose challenge (200 MID50) yielded only about 15% survival in the WIV vaccinated group while the HA and cHA groups showed approximately 80% survival. The authors use this to show heterologous protection. Using a more relevant challenge model, the authors vaccinate and challenge ferrets. In this model, HAI- and MN-antibodies were only detected in the AAV-HA-vaccinated ferrets, and all ferrets were infected with the challenge virus. Ferrets that were vaccinated with AAV-HA showed improvement after infection compared to the other vaccine groups, and the differences were mostly observed in the clinical scores, lower virus titers, and immunohistochemistry evaluation of lung sections for virus.

This study presents some interesting results on the evaluation of antibodies induced after vaccination, through evaluation of both the Fab interactions (globular head vs. stalk) and the Fc interactions. The strength of this manuscript was in the consideration of the entire antibody molecule as a mediator of protection after infection of vaccinated animals. While I find the study of interest in the area of vaccine preparation, there are a few comments that the authors need to address during the revision process.

General Comments:

1. While the antibody interactions induced were of significant interest, including the reactivity of the Fab against the globular head vs. the stalk domain, it is concerning that the antibody titers were largely undetectable using the standard correlates of protective immunity of the HAI and more recently MN assays. This was especially surprising for the WIV recipients where one would expect a robust immune response against a whole virus preparation, especially when delivered intramuscularly to ferrets. The authors should explain the potential reason for such low immunogenicity of a typically highly-immunogenic vaccine preparation, and should justify the intranasal delivery of the WIV preparation in mice rather than a more traditional intramuscular delivery. This is a critical control in the experiment, especially when the authors claim that the AAV vectors were more immunogenic. It actually appears that the WIV vaccine preparation underperformed, and/or was delivered in a manner that would not predict optimal induction of immunity,

rather than the AAV out-performed the WIV.

2. The authors performed a brief evaluation of IgA antibodies, which they claim enter the lung by transudation. This transudation will only be observed in the absence of the polymeric Ig receptor (pIgR), as described in the manuscript cited by the authors. Assuming the mice used in this study have not had the pIgR knocked out, I would assume active transport moves the IgA across the epithelial surfaces. Please see line 187 for this statement.

3. Similar to comment 2 above, is it surprising that IgA antibodies only weakly interacted with Fcγ receptors? Please see this statement on lines 440-444.

4. In Figure 2, the authors indicate that a finding of increased anti-NP antibodies in WIV recipients is unexpected. Is this really unexpected since the mice were vaccinated with WIV products (which contain NP and other common antigens) and then tested for reactivity using purified influenza viruses that were propagated in the same manner. Can the authors eliminate the potential that the broad reactivity observed is due to cross-reaction against non-HA and non-NA antigens derived from either the virus and/or eggs used to propagate the viruses? Please see lines 185-186 for this statement.

5. When vaccinating for the cHA preparations, the authors did not have individual groups for each of the cHA constructs, which may have limited the ability to generate antibodies against each of the globular heads (see Table 1). Also, since the HAI and MN titers against the H13N6 virus were relatively low, would it be possible for the authors to perform MN assays for the H2N3 and H10N7 viruses? Since the MN is a more sensitive assay there may be some antibodies that are being overlooked there.

6. It was surprising that the virus titers were high in the cHA group 14 days after Cal/7/9 challenge (Fig S5B) and in the cHA, HA, and WIV groups 14 days after high-dose PR8 (Fig S5F). This seems like a long tie for the virus to linger in the lungs of mice. Similarly, the weight loss data presented for the Ca/7/9 challenge showed an increase in weight at about Day 7 (Fig. 5B) that was then followed by a secondary decrease in weight prior to death. Can the authors explain the differences in these observations from what one might consider the normal course of an influenza virus infection in mice?

7. The high-dose and low-dose PR8 challenge values were only 2-fold different. Is there really that great of a difference between 100 MID50 and 200 MID50? What was the LD50 of the stock virus in relation to the ID50?

Specific comments

1. There were numerous times in the reading of the manuscript where the sentence structure and grammar could be improved.

2. When presenting the results of the peptide mapping study, could the authors more clearly indicate which peptides are being referred to in the text (line 242 and 243) and in the Figure (Fig S3D)? The labeling was inconsistent making the reviewer have to assume that the red bars referred to the peptides described in the text.

3. Regarding the immunohistochemistry, could the authors please let the reader know what color the anti-HA stains the sections so that we can appreciate the presence and absence of the stain?

4. On line 401 there is a superscript "58" that seems out of place.

5. In Figure S5, it appears that panel C should be panel B and vice-versa.

Point-to-point responses:

Referee #1 (Comments on Novelty/Model System for Author):

The statistical analysis is well describe in all Figures, and graphs. However, there are an important number of Figures reporting only n=2, and only two dots in the Figure! [results without SD or SEM],, and n=3 showing only 3 data points. Does this mean that no technical replicates were included in the assay? It is not appropriate to report results without technical replicates. If these results are available, then authors should show ALL data sets in the Figure.

It is not evident whether the mice experiments have been carried out with enough power calculation to assess the differences. Could the authors provide such analysis? Additionally, there are no indications whether the data represent independent experiments (ie different immunization days), or just one group of animals challenged with the candidate vaccines.

Referee #1 (Remarks for Author):

Demminger and colleagues aim to provide compelling in vitro and in vivo evidence for the use of AAV as platform to develop influenza vaccines. As proof of principle they generated AAV viruses expressing influenza virus HA or chimeric HA, and investigate the protection potential by defining the type of antibody responses, and they evaluate the protection conferred in two challenge models: mice and ferrets.

This is highly relevant topic, with huge translational potential. The use of AAV viruses is well accepted, even for vaccine development, and there are already published studies assessing its potential to generate influenza vaccines. Nevertheless, still there is a need to better understand the type of responses elicited, and how universal this platform could be. The manuscript is well written and the Figures are elegantly presented. The experimental approach is sensible, and the appropriate controls are included. I have the following comments for the authors consideration:

1. Please note my comments referring the statistical analysis, and the number of biological and technical replicates.

Re.: We do understand the concerns, but we like to assure that our analyses did include technical replicates and were conducted after a careful power analysis to be able to assess statistical differences:

a) As to the number of replicates:

We included more data points in the revised Figures Fig S1B and Fig S1G that were correctly criticized by the referee, now showing three and four independent analyses, respectively, with mean +/- SE. Please note, that addition of further experimental replicates did not change any of the conclusions, as mean values did not change largely. We also added statements as to the numbers of technical replicates into several other figure legends.

b) As to the power analysis:

To calculate the required sample size for the challenge studies described in Fig. 5 we conducted in advance an immunogenicity study and evaluated the results by using the tool G*Power 3.1.6 (University of Kiel, Germany). Serum antibody levels were measured by ELISA against Cal/7/9 of the AAV-HA group (mean 8851, SD 4298) compared to the AAV-GFP group (mean 881, SD 93). Subsequently, an a priori analysis was performed (method: t tests - Means: Wilcoxon-Mann-Whitney test (two groups), alpha set to 0.05, power set to 0.8), which calculated that the required sample size to detect this difference with sufficient power (i.e. a power of 0.8) was four animals per group. Based on this calculation we decided to use a minimum of 5 animals per group for the challenge experiments. Furthermore, we conducted a post-hoc power analysis using the data shown in Fig. S. 5 (now EV3) (maximum weight loss data, method: t tests - Means: Wilcoxon-Mann-Whitney test (two groups)) and Fig. 2b (ELISA data against Cal/7/9, method: F test - ANOVA: Fixed effects, omnibus, one-way) (see Figure A). If the calculated value is ≥ 0.8 , tests can be performed with sufficient power to assess statistical differences between groups. Taken together, these analyses demonstrate that the sample size was high enough in each group to reveal statistically significant differences of challenge and ELISA data. We inserted a sentence on these power calculations on lines 715-717 of the revised manuscript

Power Analysis Figure S. 5

Cal/7/9 challenge (Fig. S5A)

	vs. AAV-GFP			
	calculated effect size	group size	alpha	calculated power
AAV-HA	15.8452628	5	0.05	1
AAV-cHA	1.9136503	5	0.05	0.849252
WIV	2.2860132	5	0.05	0.939227
AAV-NP	2.473135	5	0.05	0.9645764

PR8 Lo challenge (Fig. S5C)

	vs. AAV-GFP			
	calculated effect size	group size	alpha	calculated power
AAV-HA	4.1451881	6	0.05	0.9999992
AAV-cHA	3.8174391	6	0.05	0.9999911
WIV	1.9510843	6	0.05	0.9205483
AAV-NP	9.3973126	6	0.05	1

PR8 Hi challenge (Fig. S5E)

	vs. AAV-GFP			
	calculated effect size	group size	alpha	calculated power
AAV-HA	1.8240024	7	0.05	0.9308491
AAV-cHA	1.8393633	7	0.05	0.9342676
WIV	0.5185967	7	0.05	0.2258195

Power Analysis Fig. 2b

	mean	overall SD	group size	calculated effect size	alpha	calculated power
AAV-HA	8314	2039	18	1.4341544	0.05	1
AAV-cHA	1824	2039	18	1.4341544	0.05	1
WIV	1635	2039	18	1.4341544	0.05	1
AAV-NP	5888	2039	11	1.4341544	0.05	1

Figure A: Post-hoc power analysis of data presented in the original manuscript in Figs. 2b and S5. Effect size was calculated based on data presented in the respective figures, group size was taken from the experiment and alpha was set to 0.05 to calculate the actually achieved power of the experiment.

c) As to the numbers of independent experiments

Each challenge experiment was done independently from the others, meaning that three different groups of mice were vaccinated and challenged at different time points. The induction of immune responses following application of the vaccines was very robust over the three challenge groups and no big inter-group variation could be observed (see Fig. 1 B and C)

2. The histology analysis do not include a more quantitative scoring. Given the comparisons of this work, this analysis should be performed in a blind way.

Re.: The histological analyses were performed by board certified veterinary pathologist (A.D.G. and K.D.) with several years of training in analysis of organ samples infected with respiratory pathogens according to standardized criteria and in a blinded fashion (Dietert, Gutbier et al., 2017). We inserted a corresponding remark in line 768 of the revised manuscript.

Thus, an unbiased analysis of the samples was possible. In the revised manuscript we included additional quantitative scoring in the new panels S6 (now EV5) H to J, which corroborates the findings described in lines 392-395 and the discussion.

3. AAV-NP viruses performed very well in the mice challenge (perhaps even better than the HA variants!), therefore it is not evident why they were excluded from the ferret experiment.

Re.: We agree that AAV-NP performed very well in mice and that it would be very interesting to evaluate its performance in ferrets as well. However, the number of animals was limited in our study, which is why we decided to focus on the comparison of the HA-based vaccines only. We indicated this on lines 334-335 of the revised manuscript.

4. A major limitation of this study is the lack of analysis of immune responses at the very least in the mouse model. There is no indication on the inflammatory parameters, as well as the recruitment of immune cells, and whether differences could be found between the AAV constructs.

Re.: Thank you for this comment. Several excellent reviews have been published describing the pathophysiological and inflammatory features of the murine influenza model, e.g. (Thangavel & Bouvier, 2014). However, it is understood in the field that the principal read-outs for protective efficacy of an intervention are: morbidity – weight loss; mortality – death at defined endpoint; viral replication – lung titers; lung pathology (Margine & Krammer, 2014). We therefore chose those parameters to assess the prophylactic efficacy of AAV-vectored vaccine towards influenza virus challenge in our proof-of-concept study. In fact, quantifications of inflammatory processes or recruitment of immune cells is rarely done at this stage of vaccine development. Our main focus was on the analysis of the humoral responses, in particular on broadly reactive antibodies induced by the different AAV-vectors against the HA protein, which is a major viral antigen. We appreciated that in future studies it will be important to evaluate differences in immune responses regarding the induction of T cells or B cells, especially including memory responses, and further to specify the exact mechanism of protection, which might include Fc-gamma receptor activated effector functions executed by e.g. NK cells. This was, however, not the scope of the current study. We had placed statements acknowledging these issues in lines 430-432, 474-476, 479-481 of the original manuscript (on lines 445-447, 491-493, 496-498 of the revised version).

5. Fig 6, why the treatment regime was different between the commercial vaccine and those under consideration in this study. Without a strict side-by-side comparison it is difficult to assess whether the new AAV vectors are indeed better.

Re.: Thank you for pointing this out. Human QIV is currently mainly an i.m. applied vaccine. In the ferret model, we wanted to directly compare what would be the standard immunization regimen in naïve humans (i.e. 2x i.m.) to our AAV-vectored i.n. vaccine, to evaluate whether the AAV-vector vaccine regimen was at least non-inferior to QIV.

6. Fig 4, although it is relevant to detect the presence of these antibodies, authors should also show whether they are functional. Assays should be performed to determine whether the antibodies elicited upregulate antiviral effect upon activation of FcγR.

Re.: We agree that analysis on the functionality of the Fc-gamma receptor activating antibody response would be informative. In recent years, studies have highlighted the importance of alveolar macrophage (He, Chen et al., 2017), neutrophil (Mullarkey, Bailey et al., 2016), and NK cell (Vanderven, Ana-Sosa-Batiz et al., 2016) effector functions in this respect. The Fc-gamma R assay used in our study has been previously validated by comparison of the activation of the reporter cell line to activation of polyclonal NK cells (via CD107a degranulation marker) by an anti-Fc-gamma receptor antibody (Corrales-Aguilar, Trilling et al., 2013) (Fig. 5 therein). Both, activation of the reporter cell line and NK cells correlated well, demonstrating that the reporter assay accurately reflects activation of an antiviral effector functions. Thus, we think that further demonstration of exerted antibody effector functions is out of scope of this manuscript, especially as measurement of Fc-gamma R activation seems to be widely accepted as correlates of Fc-gamma R function (Impagliazzo, Milder et al., 2015, Tan, Leon et al., 2016, Wohlbold, Podolsky et al., 2017, Yasuhara, Yamayoshi et al., 2019).

Referee #2 (Comments on Novelty/Model System for Author):

Except for the medical impact, I consider the experimental setups of high technical quality. I do have reservation on the eventual clinical applicability of an AAV vectored influenza vaccine.

Referee #2 (Remarks for Author):

Demming and co-authors report on the generation, immunogenicity and protective potential of a set of AAV vectors that can express influenza A hemagglutinin and derivatives thereof. Wild type, chimeric as well as stalk-only constructs based on Cal/09 HA were cloned into the AAV vector. Transduction of 293T cells at a MOI of 10⁶ resulted in detectable expression of all constructs by Westernblot. The headless HA constructs displayed poor reactivity with C179. In mice, 3 intranasal immunizations with AAV-HA result in higher Cal/09 binding and neutralizing serum IgG antibodies than two intranasal doses of WIV. HI and MN titers in serum of immunized mice were only detected in AAV-HA and AAV-H13H1 chimeric HA. Mouse immune sera from AAV-HA, AAV-cHA and WIV could bind to denatured HA2 of H1N1 viruses that were isolated between 1918 and 2009. AAV-HA induced both head- and stalk-specific antibodies; WIV and AAV-cHA primarily stalk-specific antibodies. An in vitro cell-based FcγR-activation assay allowed to conclude that AAV-HA immune serum induced the strongest FcγR activation against Cal/09 infected cells and, although much weaker, against PR8 infected cells. In line with the poor immunogenicity of WIV in this model, WIV immune serum had a very weak capacity to activate FcγRs. FcγR activating antibodies in the immune sera were directed against HA head and stalk for the cHA vaccine and primarily directed against the HA head for the AAV-HA vaccine. AAV-HA immunized mice were best protected against a homologous challenge. Interestingly, the mouse that had to be euthanized in the challenged WIV group had no remaining virus in the lungs at the endpoint. AAV-HA and -cHA protect fully against a dose of PR8 virus challenge and partially against challenge with a double dose of PR8.

The protective potential of 3 intranasal immunizations with AAV-HA and -cHA was largely recapitulated in a ferret model. Protection against homologous challenge was clearest in 3 times AAV-HA immunized ferrets, in line with the observation that only in this group neutralizing antibodies were detected.

Overall, the experiments are very well performed and quite novel. The applicability of 3 intranasal immunizations with AAV-vectored influenza HA for seasonal influenza in human is not trivial.

Major remarks:

1. The statement in line 319 is misleading. The effect on survival of HA-specific FcγR-activating antibodies was not evaluated in the experiment. The immunized mice had likely also mounted T cell responses against the recombinant vaccine antigens and those responses may have contributed to protection. A passive serum transfer experiment followed by challenge, preferentially in wt and FcγR deficient mice could reveal an effect of FcγR-activating antibodies on protection.

Re.: We agree that the statement can be misleading since we cannot rule out that T cells might have contributed to protection. We changed the sentence to “To evaluate the effect of the HA-based AAV-vector vaccine on protection, we conducted ...” (now on lines 334-335).

2. An AAV-NP group should be included in Fig. 5C to support the conclusion that AAV-NP is superior to WIV in protection against a heterologous challenge as stated in line 328. In addition the statement should explicitly mention that 3 immunizations are compared with 2 immunizations. Does the statement on the comparison with WIV hold when protection after 2 immunizations with AAV-HA is evaluated?

Re.: Thank you for this comment. The AAV-NP vaccine was in fact not evaluated with the higher dose of PR8 and we therefore have not a complete dataset for this vector. To match the criticism, we changed the statement in line 328 to “In summary, these results indicate that three doses of AAV-HA or AAV-cHA were superior to two doses of WIV in reducing mortality and disease severity against heterologous influenza challenge in mice” (now on lines 342-344 of the revised manuscript). We also eliminated mentioning of the AAV-NP at the end of the introduction for the same reason in lines 123-125.

We have currently not tested the minimally required number of AAV-vector immunization that suffices to induce protective immunity comparable or superior to WIV vaccination. That is because the suggested mode of action of the chimeric HA constructs necessitated three immunizations in order to be able to boost the antibody response against the HA stalk (Krammer, Pica et al., 2013, Margine, Krammer et al., 2013). To allow for a head-to-head comparison to AAV-cHA, also for AAV-HA and AAV-NP three immunizations were chosen. Compared to WIV, data on antibody responses after the first and second immunization show a rapid and strong increase of AAV-HA and AAV-NP induced antibodies, suggesting that two immunizations might suffice to induce a protective immunity (see Figure B). However, this expectation will need to be tested in future work.

Figure B: Antibody responses in mice three weeks after each immunization. AAV-HA, AAV-cHA and AAV-NP received three immunizations, WIV received two immunizations. Blood was taken from each animal at the indicated time point after the immunizations. Serum samples were pooled per vaccine group and binding to the homologous Cal/7/9 virus was analysed by ELISA (mean +/- SEM).

3. The conclusion in line 506: AAV-vector mitigates immunodominance of strain-specific epitopes in the HA head does not appear to be supported by the data. Figures 3A and 4C (left hand panels) show a predominant response against HA1 and full length HA, respectively. A comparison with a setting in mouse or ferrets in which the WIV or QIV vaccine had actually worked, would also be needed to support the statement. I.m. injection of adjuvanted WIV or QIV will typically results in clear seroconversion in mouse and ferret models. Intranasal WIV adjuvanted with polyI,C will likely also result in a better seroconversion.

Re.: We understand the criticism and have eliminated the statement on the mitigation of immunodominance by AAV-vectors in line 529 of the amended manuscript. This is in part due to results of our additional immunization analysis that we conducted in response to this comment (shown in Fig. S2 of the amended manuscript). Here, we observed that the intranasal delivery of WIV alone altered the binding spectrum of HA-specific antibodies in comparison to traditional i.m. immunization with WIV, which had HAI⁺ and MN⁺ activity and recognized also HA1 (Fig. S2), which in other words, in fact, “worked”. This is an interesting observation that deserves further attention, but which is out of the scope of this manuscript. Hence, more comprehensive studies possibly also involving the analysis of germinal center B cells will be needed (Angeletti, Kosik et al., 2019) to assess the aspect of epitope specificity generated by AAV vectors. In the literature mainly HA-stalk antibodies are being considered as activators of Fc-gammaR functions, though more recent publications showed that also antibodies against the head can induce Fc-gammaR, as long as they not interfere with receptor binding of the HA (Cox, Kwaks et al., 2016, Leon, He et al., 2016). Significantly, our results show, that AAV-HA is able to induce Fc-gammaR activating antibodies targeting both the HA-stalk and head domains, therefore representing an advantage over vaccines that induce such antibodies “only” against the HA-stalk such as headless HA proteins or chimeric HAs. This is most likely a consequence of the specific way of antigen presentation, which is mediated only by the AAV-vector, but not by the inactivated vaccine.

4. Figure 4C: which FcγR(s) was tested with the transfected cells as target? Please clarify in the legend.

Re.: The Fc-gamma-receptor 1 was used in the experiment depicted in Fig. 4C due to the following reasons: This receptor is one of the murine activating Fc-gamma-receptors and the activity of AAV-HA and AAV-cHA serum against whole Cal/7/9 virus were more comparable against this receptor in comparison to the other activating receptors (see Fig. 4A, left panel), allowing a better evaluation of the proportions of Fc-gamma-receptor activating antibodies against the full length vs. stalk-only HA in the AAV-cHA and AAV-HA sera. We have added this information to the caption of Figure 4.

5. The authors should discuss the limitations of the production capacity and downstream processing steps that would be required to prepare millions of doses on an annual basis of a quadrivalent AAV-vectored influenza vaccine. In addition, there is a potential safety concern. An important step in the purification process of biologicals, such as recombinant monoclonal antibodies, is a virus removal step that involves ultrafiltration. It would be interesting for the reader if the authors could address these concerns in the discussion.

Re.: We are aware that production capacity for AAV-vectors is currently limited, but is at least sufficient for the production of batches suitable for mid-size clinical studies (<https://clinicaltrials.gov/ct2/results?cond=&term=aav&cntry=&state=&city=&dist=>). Although the minimally required dose for an AAV-vector influenza vaccine will need to be defined, current up-scalable production platforms will likely need further development to yield pure and potent AAV-vector product to titers high enough titers to be used for mass vaccination. However, most promising appear the use of Baculovirus

expression system for AAV-vector production together with downstream purification steps including affinity and ion exchange chromatography without a requirement for ultrafiltration (Nass, Mattingly et al., 2017). We added a statement addressing the issue on lines 523 – 526 of the amended manuscript.

Other remarks:

1. Please clarify why expression of mHL1 and mHL2 is detectable in the cytoplasm (Figure 1C and D). Expression in the cell culture medium should also be documented for these 2 constructs, which lack a TM domain.

Re.: In the analysis shown in Fig. 1C, cells were transfected with the AAV-constructs, fixed, permeabilized and stained with specific antibodies. Hence, proteins inside the cell (including those in secretory compartments) and membrane proteins can be detected with this method. In the SDS-PAGE/WB analysis shown in Fig.1D, complete cell lysates of AAV-vector transduced cells were prepared using RIPA buffer after removal of cell culture medium and washing of the cells. Thus, as in Fig. 1C, intracellular and membrane proteins can be detected. It is therefore not surprisingly, that mHL1 and mHL2 can be detected with the two methods. In addressing the issue we compared expression of the two constructs in transfected cells and the corresponding culture supernatants in Fig C (see below). The analysis showed relatively strong signals for both stem constructs in the cell lysates with relative stronger detection of mHL2 than mHL1 (Fig. C, lower panel). Both constructs were also detected in the SUP, with mHL1 apparently being efficiently released, whereas only low amounts were stained for mHL2 suggesting inefficient release (Fig. C, upper panel). For the latter analysis, 20 µl of cell culture supernatant (i.e. of in total 1 ml culture medium per 12-well) were separated under denaturing and reducing conditions via SDS-PAGE, transferred to nitrocellulose membrane and detected with anti-V5 tag antibody, suggesting that the amount of protein released by the culture is quite high.

Figure C: Immunoblot analysis of mHL1 and mHL2 in cell culture supernatant and cell lysates, respectively. Cell culture supernatants or cell lysates of 293T cell 48 hours after transfection with the indicated AAV constructs were separated under reducing and denaturing conditions via SDS-PAGE, blotted onto nitrocellulose membranes and detected with an anti-V5 tag antibody.

2. Line 152: at a MOI of 1 million, it is questionable that the transduction rate can be considered as high. Please modify the statement. In addition, a dose titration range (different MOIs) should be performed to convincingly show that the different HA variant vectors can give rise to comparable expression levels based on V5-tag reactivity. The transduction experiment of which a Western Blot is shown in Figure 1D, should be better described: in line 574, a plasmid transfection is described, not an AAV transduction.

Re.: Thank you for pointing out that Fig. 1D analyzed transduced cells. We added in the revised manuscript a corresponding sentence in lines 599-601 in the method section. We agree that the term “high” may overstate the transduction rate of the AAV-vector preparations. However, given that most AAV-vector serotypes (including AAV9) show only moderate to low transduction efficiency *in vitro* (Ellis, Hirsch et al., 2013), the AAV-vectors generated in our study match the expectations regarding transduction rate. We have thus changed the statement to “... robust...” to indicate that antigen expression by the transduced vectors was easily detectable in our experiments.

As shown in Fig 1D and Fig S1G, expression levels and transduction rates of AAV-cHA vectors were slightly lower compared to the other AAV-vectors. We are aware that the amount of protein that is produced after transduction will influence immunogenicity, which is why a high transduction rate is desirable. However, the scope of our study was not to compare the antigens per se in a quantitative manner (which was already done in the original studies describing the antigens (Krammer et al., 2013, Margine et al., 2013), but to qualitatively evaluate the feasibility of AAV-vectors to serve as vehicles for these antigens. The experiment shown in Fig. E of this rebuttle (addressing criticism of referee 3) showed that an alteration of the cHA immunization scheme, in which cHA1 was given first (instead of cHA3 shown in the manuscript), resulted in overall comparable MN titers against the prime subtype HA. This suggests that the AAV vectors at least stimulated comparable humoral responses in the vaccinated mice. This interpretation is also supported by the blot data shown in Fig. 3 that allows a direct comparison of the antibodies reactivities induced by the different vaccines.

3. Line 1084:"Statistical ..." belongs to the legend of 2B.

Re.: Agreed. We have corrected the legend of Fig. 2B.

4. Please add a statement on the number of repeat experiments that were performed in the legend of Figure 5 (mouse immunization and challenge).

Re.: We have extended the statement on the number of animals used in the panels and now provide information on the numbers of animal for each challenge group in the legend to Fig.5.

5. Line 712 and table S1: what is the correct dose that was used for the ferret immunizations?

Re.: We apologize for this mistake, apparently the mouse AAV-vector dose was falsely copied to the rows of the table describing the ferret study. The table now shows the correct number (7.5×10^{12} GC) that is also mentioned in line 712 in the original manuscript (now on line 740).

6. Line 671: the correct reference for the FcgR activation assay is Corrales-Aguilar et al., JIM 2013.

Re.: We agree that Corrales-Aguilar et al. were the first to describe and characterize the human Fc-gamma receptor assay. However, the reference that is mentioned in our manuscript describes/uses the murine Fc-gamma receptor assay for the first time, which was generated based on the human assay by Katrin Ehrhardt, a co-author of the van den Hoecke study, who also provided us with the cell lines. Thus, we now included both references in line 671 (now on lines 696-697).

Referee #3 (Remarks for Author):

Comments for the authors of the EMBO Molecular Medicine manuscript number EMM-2019-10938:

The authors of the EMBO Molecular Medicine manuscript "Adeno-associated virus-vectored influenza vaccine elicits neutralizing and Fc γ receptor-activating antibodies", present interesting findings in their work with adeno-associated virus (AAV) vectors for vaccination against influenza viruses. Specifically, the authors used AAV technology to express full-length influenza virus hemagglutinin (HA), as well as chimeric HA constructs that included the globular head from H2 influenza viruses, H10 influenza viruses, or H13 influenza viruses. All of these constructs contained the stalk from the pandemic H1N1 virus A/California/7/2009. The authors also expressed headless and modified headless domains of the A/California/7/2009 stalk. As a candidate for a highly conserved epitope the authors also expressed the influenza A virus nucleoprotein (NP) within the context of the AAV vector. The authors show vaccine-induced immunity against H1N1 and H5N1 influenza viruses after vaccination with the full-length HA and immunity against Group 1 viruses (H1, H2, H5, and H13 HAs) but not group 2 viruses (H7 and H3 HAs) after vaccination with the cHAs (delivered a single time each sequentially with H13 first, H2 second, and H10 third, rather than as individual groups). Immunity against the NP recognized viruses from both groups as did antibodies against the whole influenza virus (WIV) vaccine. While antibodies were detected in these various groups, antibodies that inhibited hemagglutination (hemagglutination inhibition, HAI) and neutralized virus infectivity (microneutralization, MN) were only induced against the A/California/7/2009 virus in the AAV-HA vaccine group and against the H13N6 virus in the AAV-cHA group. Further evaluation of the antibody response identified an interaction with the HA2 domain after vaccination with AAV-HA, AAV-cHA, and WIV. These antibodies also activated Fc receptors, specifically the RI, RIII, and RIV receptors that are involved in ADCC reactions. Anti-NP antibodies (induced by the AAV-NP vaccine) reacted with Fc receptors from the RI, RII, RIII, and RIV classes.

When tested in a challenge model with A/California/7/2009, the HA and NP vaccines protected 100% of mice from death, while the cHA and WIV vaccines protected about 80% of mice. When PR8 was used for challenge, the low-dose challenge (100 MID50) did not induce any death while a high-dose challenge (200 MID50) yielded only about 15% survival in the WIV vaccinated group while the HA and cHA groups showed approximately 80% survival. The authors use this to show heterologous protection. Using a more relevant challenge model, the authors vaccinate and challenge ferrets. In this model, HAI- and MN-antibodies were only detected in the AAV-HA-vaccinated ferrets, and all ferrets were infected with the challenge virus. Ferrets that were vaccinated with AAV-HA showed improvement after infection compared to the other vaccine groups, and the differences were mostly observed in the clinical scores, lower virus titers, and immunohistochemistry evaluation of lung sections for virus.

This study presents some interesting results on the evaluation of antibodies induced after vaccination, through evaluation of both the Fab interactions (globular head vs. stalk) and the Fc interactions. The strength of this manuscript was in the consideration of the entire antibody molecule as a mediator of protection after infection of vaccinated animals. While I find the study of interest in the area of vaccine preparation, there are a few comments that the authors need to address during the revision process.

General Comments:

1. While the antibody interactions induced were of significant interest, including the reactivity of the Fab against the globular head vs. the stalk domain, it is concerning that the antibody titers were largely undetectable using the standard correlates of protective immunity of the HAI and more recently MN assays. This was especially surprising for the WIV recipients where one would expect a robust immune response against a whole virus preparation, especially when delivered intramuscularly to ferrets. The authors should explain the potential reason for such low immunogenicity of a typically highly-immunogenic vaccine preparation, and should justify the intranasal delivery of the WIV preparation in mice rather than a more traditional intramuscular delivery. This is a critical control in the experiment, especially when the authors claim that the AAV vectors were more immunogenic. It actually appears that the WIV vaccine preparation under-performed, and/or was delivered in a manner that would not predict optimal induction of immunity, rather than the AAV out-performed the WIV.

Re.: We are happy to address this conceptually important question. It appears counterintuitive at first glance that WIV did not readily induce homologous HAI or MN antibody responses. To address the issue, we performed in mice an immunization experiment to rule out that there was a quality problem with the killed vaccine, which could explain an "underperformance". The results of this analysis are shown in the new Fig S2. Here we applied into mice the same WIV preparation and dosings as were used throughout the original manuscript, using either the intranasal (used in our original study) or intramuscular routes. As expected, i.m. vaccination led to the induction of easily detectable HAI⁺ and MN⁺ antibodies confirming that the killed vaccine used was in fact highly immunogenic (Fig. S2B, C). Delivery by the i.n. route, in contrast, induced influenza-specific IgG, but did not elicit HAI or MN antibodies. This was in fact not completely unexpected, as Bhide et al. had previously observed that induction of HAI⁺ and MN⁺ antibodies by WIV *via* the i.n. route required adjuvantation (Bhide, Dong et al., 2019). We described this in the original manuscript in lines 437-440 (now on lines 452-455). It was a remarkable finding in the study by Bhide et al. that i.n. WIV immunization even without adjuvantation still

protected mice completely against a homologous viral challenge, and conferred protection against a heterologous H1N1 challenge very similar to WIV given the i.m. route. In our study, we wanted to directly compare immune responses induced by the AAV and the inactivated vaccine, and therefore chose for both classes the same route of delivery. Considering the published data (Bhide et al.; 2019), we believe that WIV given the i.n. route was a reasonable standard to test against.

Ferrets are considered the animal model recapitulating most aspects of human influenza. The vaccine preparation we used in our study was a non-adjuvanted commercial human quadrivalent inactivated vaccine (QIV). The two doses of QIV vaccine were given as the best available control, although we are aware that such inactivated vaccine may not be highly immunogenic in naïve ferrets without adjuvants (Baras, de Waal et al., 2011), as we mentioned in the original manuscript on lines 496 – 498. To indicate this fact more clearly we complemented this statement on lines 513–515 of the revised manuscript. Still, as can be appreciated from the histological staining shown in Fig 6 G and H, there was some degree of protection from QIV compared to AAV-GFP and AAV-cHA even in the absence of neutralizing antibodies.

2. The authors performed a brief evaluation of IgA antibodies, which they claim enter the lung by transudation. This transudation will only be observed in the absence of the polymeric Ig receptor (pIgR), as described in the manuscript cited by the authors. Assuming the mice used in this study have not had the pIgR knocked out, I would assume active transport moves the IgA across the epithelial surfaces. Please see line 187 for this statement.

Re.: We apologize for this improper statement. We agree that active transport of IgA across the mucosal epithelium is mainly responsible for their abundance in the lung of wildtype mice. We corrected the statement on lines 194 – 195 of the revised manuscript accordingly to read now “...due to their high local abundance in the airway mucosa.”

3. Similar to comment 2 above, is it surprising that IgA antibodies only weakly interacted with Fcγ receptors? Please see this statement on lines 440-444.

Re.: Agreed, this sentence could have been misunderstood. We agree that it is not surprising that IgA do not activate Fc-gamma R. We rephrased the paragraph in lines 452 – 459 of the amended manuscript to more accurately present our argumentation.

4. In Figure 2, the authors indicate that a finding of increased anti-NP antibodies in WIV recipients is unexpected. Is this really unexpected since the mice were vaccinated with WIV products (which contain NP and other common antigens) and then tested for reactivity using purified influenza viruses that were propagated in the same manner. Can the authors eliminate the potential that the broad reactivity observed is due to cross-reaction against non-HA and non-NA antigens derived from either the virus and/or eggs used to propagate the viruses? Please see lines 185-186 for this statement.

Re.: We cannot completely exclude contributions of non-HA and non-NA antibodies to the apparent broad reactivity. However, WIV vaccination did at least not induce substantial levels of anti-NP antibodies, which can be appreciated from the Western Blot in Figure 3. Also, we performed additional immunofluorescence stainings with WIV serum against transfected NP protein and did not observe specific signals (see Figure D). The Western Blot in Figure 3 also indicates that none of the other more abundant influenza proteins (e.g. M1) led to an induction of an immune response. Thus, the immune response seems to largely target HA. We agree that the statement in lines 185 – 186 of the original manuscript could be misleading and changed the statement to “Unexpectedly, WIV vaccination also induced broadly reactive antibodies, covering several viral subtypes of group 1 and 2 a, though at lower intensities” (in lines 192-193 of the revised manuscript).

Figure D: Immunofluorescence staining of cells transfected with NP protein with mouse serum induced by AAV-HA, WIV or AAV-NP vaccines

5. When vaccinating for the cHA preparations, the authors did not have individual groups for each of the cHA constructs, which may have limited the ability to generate antibodies against each of the globular heads (see Table 1). Also, since the HAI and MN titers against the H13N6 virus were relatively low, would it be possible for the authors to perform MN assays for the H2N3 and H10N7 viruses? Since the MN is a more sensitive assay there may be some antibodies that are being overlooked there.

Re.: The goal of using the cHA was not to induce HAI or MN antibodies against the divergent head regions of each cHA protein, but to boost antibodies against the conserved HA-stalk domain. This can only be achieved when the same HA-stalk domain is repeatedly presented to the immune system while the head domains change, as is the case during a sequential immunization with different cHA proteins. The lack of HAI and MN antibodies against the H2 and H10 HA suggests that a re-focusing of the antibody response from the head domain towards the stalk domain did occur in our study during the sequential immunization with the cHA proteins. Due to the lack of remaining serum from these animals it is unfortunately not possible to perform further serological analysis. However, we performed a pilot immunogenicity analysis in advance of the presented study for which HAI and MN titers are available (see Figure E). In this pilot study we used a different order of the cHA for immunization (cHA1 (H2 Head) – cHA3 (H13 head) – cHA2 (H10 head)). Although the order of the cHA was different, we observed the same pattern as in the study described in the manuscript: HAI⁺ and MN⁺ antibodies were induced only against the head of the HA used for prime, i.e. H2, but not against the other two HA. This indicates that the characteristic pattern of antibody responses we observed in our studies is due to the specific activation of the immune system by a sequence of cHA and not due to a technical limitation when determining MN antibodies for one of the other subtypes. We did not include this dataset into the manuscript as animals vaccinated with this scheme were not further analysed.

Figure E: MN antibody induction against vaccine strains (pilot immunogenicity study).

Mice were vaccinated according to the scheme depicted. Terminal serum samples were used to determine MN antibody responses against the indicated viruses.

6. It was surprising that the virus titers were high in the cHA group 14 days after Cal/7/9 challenge (Fig S5B) and in the cHA, HA, and WIV groups 14 days after high-dose PR8 (Fig S5F). This seems like a long tie for the virus to linger in the lungs of mice. Similarly, the weight loss data presented for the Ca/7/9 challenge showed an increase in weight at about Day 7 (Fig. 5B) that was then followed by a secondary decrease in weight prior to death. Can the authors explain the differences in these observations from what one might consider the normal course of an influenza virus infection in mice?

Re.: Yes, we are happy to clarify these aspects.

a) As to the lung virus loads:

Panels B, D and F (right side) in previous Figure S5 (now EV4) include all animals from each vaccine group, i.e. animals succumbing to the infection and reaching the individual endpoint during the 14-day monitoring period, as well as protected animals that survived the complete challenge period of 14 days. This is why the x-axis is labelled 'endpoint/ day 14'. Only Figure S5F (now EV4) (left side) is showing animals that were all euthanized on day 3. To present these settings in a more visible way we included an additional color coding in the previous Fig. S5 (now EV4) B, D, F showing which animal survived up to day 14 and which one did not (day 14 = black, endpoint = red). This emphasizes that animals succumbing to the infection had high lung virus loads on the day they had to be euthanized, while animals surviving up to day 14 were – as expected and correctly pointed out by the referee – able to clear the virus from their lungs during that period.

b) As to the weight loss data:

Thank you for this vigilant remark. The weight curves for the groups AAV-HA, AAV-cHA, AAV-NP and WIV appear to be according to expectations, and variations in the course of the curve are due to biological variation between the animals per each group. Very likely, the referee is thus specifically referring to the AAV-GFP group. As shown in Fig. 5A, one AAV-GFP immunized animals survived up to day 8, while all other animals succumb to the infection on day 4. A peak in the weight loss curve can be seen at day 6 post infection and not at day 5, as would be expected, when the weight loss curves of the four euthanized animals end at this day and the remaining curve contains only data from the one animal surviving up to day 8 (see correct curves of individual animals in Figure F). While revising the manuscript we found that there was an error presentation of the data in Fig 5 B, as weight data of the AAV-GFP animals dying at day 4 was falsely copied to the day 5, explaining this peculiar weight curve. We apologize for this mistake and corrected Figure 5 B accordingly, but the changes did not alter any of the conclusions drawn: None of the AAV-GFP immunized animals was protected against the homologous challenge, although one out of four animals succumbed slower to the infection.

Figure F: Weight curves of individual animals of the AAV-GFP group during Cal/7/9 challenge. The mean of the weight curves, which is shown in Fig. 5 B of the manuscript, is overlaid as dotted line.

7. The high-dose and low-dose PR8 challenge values were only 2-fold different. Is there really that great of a difference between 100 MID50 and 200 MID50? What was the LD50 of the stock virus in relation to the ID50?

Re.: The aim of increasing the challenge virus dose was to resolve differences in the protective capacity of the AAV-vector vaccines compared to WIV towards survival, as the low dose challenge could only show that WIV immunized animals lost more weight than to the AAV-vector immunized groups (Fig. 5 D). Vice versa, a challenge dose too high would have artificially concealed protective effects of the vaccines. For this purpose a two-fold increase of the challenge dose was appropriate (Fig. 5 E). Unfortunately, we only have mouse infectious dose 50 (MID50) titrations available for the virus stocks used in the study, which is stated in the methods section (lines 724-725).

Specific comments

1. There were numerous times in the reading of the manuscript where the sentence structure and grammar could be improved.

Re.: We thoroughly screened the manuscript for such shortcomings and amended them throughout.

2. When presenting the results of the peptide mapping study, could the authors more clearly indicate which peptides are being referred to in the text (line 242 and 243) and in the Figure (Fig S3D)? The labeling was inconsistent making the reviewer have to assume that the red bars referred to the peptides described in the text.

Re.: We agree that the lack of labels for the peptides makes grasping the information unnecessary hard. We thus included labels of the peptides in lines 254 – 255 (revised version) matching those in Fig S3D (now Fig. EV2D).

3. Regarding the immunohistochemistry, could the authors please let the reader know what color the anti-HA stains the sections so that we can appreciate the presence and absence of the stain?

Re.: The anti-HA primary antibody was detected with a secondary antibody conjugated to alkaline phosphatase. Binding was detected with new fuchsin, yielding a reddish staining. A statement has been introduced into the respective Materials and Methods section and the caption to Fig. 6 to indicate this.

4. On line 401 there is a superscript "58" that seems out of place.

Re.: Agreed, we deleted this number.

5. In Figure S5, it appears that panel C should be panel B and vice-versa.

Re. Thank you. We carefully double-checked the referencing of these Figure panels in the main text as well as description of the panels in the caption to Figure S5 (now Fig EV4). We believe that they appear to be correct giving no reason for a change.

Cited references:

- Angeletti D, Kosik I, Santos JJS, Yewdell WT, Boudreau CM, Mallajosyula VVA, Mankowski MC, Chambers M, Prabhakaran M, Hickman HD, McDermott AB, Alter G, Chaudhuri J, Yewdell JW (2019) Outflanking immunodominance to target subdominant broadly neutralizing epitopes. *Proc Natl Acad Sci U S A*
- Baras B, de Waal L, Stittelaar KJ, Jacob V, Giannini S, Kroeze EJ, van den Brand JM, van Amerongen G, Simon JH, Hanon E, Mossman SP, Osterhaus AD (2011) Pandemic H1N1 vaccine requires the use of an adjuvant to protect against challenge in naive ferrets. *Vaccine* 29: 2120-6
- Bhide Y, Dong W, Gribonika I, Voshart D, Meijerhof T, de Vries-Idema J, Norley S, Guilfoyle K, Skeldon S, Engelhardt OG, Boon L, Christensen D, Lycke N, Huckriede A (2019) Cross-Protective Potential and Protection-Relevant Immune Mechanisms of Whole Inactivated Influenza Virus Vaccines Are Determined by Adjuvants and Route of Immunization. *Frontiers in immunology* 10
- Corrales-Aguilar E, Trilling M, Reinhard H, Merce-Maldonado E, Widera M, Schaal H, Zimmermann A, Mandelboim O, Hengel H (2013) A novel assay for detecting virus-specific antibodies triggering activation of Fcγ receptors. *J Immunol Methods* 387: 21-35
- Cox F, Kwaks T, Brandenburg B, Koldijk MH, Klaren V, Smal B, Korse HJ, Geelen E, Tettero L, Zuijdgeest D, Stoop EJ, Saeland E, Vogels R, Friesen RH, Koudstaal W, Goudsmit J (2016) HA Antibody-Mediated FcγRIIIa Activity Is Both Dependent on FcR Engagement and Interactions between HA and Sialic Acids. *Frontiers in immunology* 7: 399
- Dietert K, Gutbier B, Wienhold SM, Reppe K, Jiang X, Yao L, Chaput C, Naujoks J, Brack M, Kupke A, Peteranderl C, Becker S, von Lachner C, Baal N, Slevogt H, Hocke AC, Witzernath M, Opitz B, Herold S, Hackstein H et al. (2017) Spectrum of pathogen- and model-specific histopathologies in mouse models of acute pneumonia. *PLoS one* 12: e0188251
- Ellis BL, Hirsch ML, Barker JC, Connelly JP, Steininger RJ, 3rd, Porteus MH (2013) A survey of ex vivo/in vitro transduction efficiency of mammalian primary cells and cell lines with Nine natural adeno-associated virus (AAV1-9) and one engineered adeno-associated virus serotype. *Virology journal* 10: 74-74
- He W, Chen CJ, Mullarkey CE, Hamilton JR, Wong CK, Leon PE, Uccellini MB, Chromikova V, Henry C, Hoffman KW, Lim JK, Wilson PC, Miller MS, Krammer F, Palese P, Tan GS (2017) Alveolar macrophages are critical for broadly-reactive antibody-mediated protection against influenza A virus in mice. *Nature communications* 8: 846
- Impagliazzo A, Milder F, Kuipers H, Wagner MV, Zhu XY, Hoffman RMB, van Meersbergen R, Huizingh J, Wanningen P, Verspuij J, de Man M, Ding ZQ, Apetri A, Kukrer B, Sneekes-Vriese E, Tomkiewicz D, Laursen NS, Lee PS, Zakrzewska A, Dekking L et al. (2015) A stable trimeric influenza hemagglutinin stem as a broadly protective immunogen. *Science (New York, NY)* 349: 1301-1306
- Krammer F, Pica N, Hai R, Margine I, Palese P (2013) Chimeric hemagglutinin influenza virus vaccine constructs elicit broadly protective stalk-specific antibodies. *J Virol* 87: 6542-50
- Leon PE, He W, Mullarkey CE, Bailey MJ, Miller MS, Krammer F, Palese P, Tan GS (2016) Optimal activation of Fc-mediated effector functions by influenza virus hemagglutinin antibodies requires two points of contact. *Proceedings of the National Academy of Sciences of the United States of America* 113: E5944-E5951
- Margine I, Krammer F (2014) Animal models for influenza viruses: implications for universal vaccine development. *Pathogens* 3: 845-74
- Margine I, Krammer F, Hai R, Heaton NS, Tan GS, Andrews SA, Runstadler JA, Wilson PC, Albrecht RA, Garcia-Sastre A, Palese P (2013) Hemagglutinin stalk-based universal vaccine constructs protect against group 2 influenza A viruses. *J Virol* 87: 10435-46

Mullarkey CE, Bailey MJ, Golubeva DA, Tan GS, Nachbagauer R, He W, Novakowski KE, Bowdish DM, Miller MS, Palese P (2016) Broadly Neutralizing Hemagglutinin Stalk-Specific Antibodies Induce Potent Phagocytosis of Immune Complexes by Neutrophils in an Fc-Dependent Manner. *mBio* 7

Nass SA, Mattingly MA, Woodcock DA, Burnham BL, Ardinger JA, Osmond SE, Frederick AM, Scaria A, Cheng SH, O'Riordan CR (2017) Universal Method for the Purification of Recombinant AAV Vectors of Differing Serotypes. *Molecular therapy Methods & clinical development* 9: 33-46

Tan GS, Leon PE, Albrecht RA, Margine I, Hirsh A, Bahl J, Krammer F (2016) Broadly-Reactive Neutralizing and Non-neutralizing Antibodies Directed against the H7 Influenza Virus Hemagglutinin Reveal Divergent Mechanisms of Protection. *PLoS pathogens* 12: e1005578

Thangavel RR, Bouvier NM (2014) Animal models for influenza virus pathogenesis, transmission, and immunology. *J Immunol Methods* 410: 60-79

Vandervan HA, Ana-Sosa-Batiz F, Jegaskanda S, Rockman S, Laurie K, Barr I, Chen W, Wines B, Hogarth PM, Lambe T, Gilbert SC, Parsons MS, Kent SJ (2016) What Lies Beneath: Antibody Dependent Natural Killer Cell Activation by Antibodies to Internal Influenza Virus Proteins. *EBioMedicine* 8: 277-290

Wohlbold TJ, Podolsky KA, Chromikova V, Kirkpatrick E, Falconieri V, Meade P, Amanat F, Tan J, tenOever BR, Tan GS, Subramaniam S, Palese P, Krammer F (2017) Broadly protective murine monoclonal antibodies against influenza B virus target highly conserved neuraminidase epitopes. *Nature microbiology* 2: 1415-1424

Yasuhara A, Yamayoshi S, Kiso M, Sakai-Tagawa Y, Koga M, Adachi E, Kikuchi T, Wang IH, Yamada S, Kawaoka Y (2019) Antigenic drift originating from changes to the lateral surface of the neuraminidase head of influenza A virus. *Nature microbiology* 4: 1024-1034

2nd Editorial Decision

23rd Jan 2020

Thank you for the submission of your revised manuscript to EMBO Molecular Medicine. We have now received the enclosed reports from the referees who were asked to re-assess it. As you will see, the reviewers are now globally supportive and I am pleased to inform you that we will be able to accept your manuscript pending editorial final amendments.

***** Reviewer's comments *****

Referee #1 (Remarks for Author):

I commend the authors for their efforts to meet the issues raised by this reviewer (as well as the others). The manuscript has certainly improved. It is unfortunate that the authors have chosen not to meet my comment referring to assessing the immune populations. Still I strongly believe this is the type of high quality pre clinical data that should be obtained in this type of work. I believe that the lack of this information may delay any potential translational impact. The infection parameters presented, while potentially meaningful in the field, are still very crude and fall short of in-depth mechanistic scrutiny. However, having said this, I also appreciate that the strength of the other evidence is much more solid in the present version of the study and do deserve to be exposed upon publication to the scientific community.

Referee #3 (Remarks for Author):

Thank you for addressing the previous comments. I have no further suggestions at this time.

2nd Revision - authors' response

11th Feb 2020

The authors performed the requested editorial changes.

Referee #1 (Remarks for Author):

I commend the authors for their efforts to meet the issues raised by this reviewer (as well as the others). The manuscript has certainly improved. It is unfortunate that the authors have chosen not to meet my comment referring to assessing the immune populations. Still I strongly believe this is the type of high quality pre-clinical data that should be obtained in this type of work. I believe that the lack of this information may delay any potential translational impact. The infection parameters presented, while potentially meaningful in the field, are still very crude and fall short of in-depth mechanistic scrutiny. However, having said this, I also appreciate that the strength of the other evidence is much more solid in the present version of the study and do deserve to be exposed upon publication to the scientific community.

Re.: We highly appreciate the comment. As written in the previous rebuttal letter we totally agree that it is absolutely important in future work to evaluate immune responses regarding the engagement of immune cells as well as inflammatory parameters to specify the main mechanism(s) of protection.

Referee #3 (Remarks for Author):

Thank you for addressing the previous comments. I have no further suggestions at this time.

Re.: We like to thank for the comment.

Corresponding Author Name: Thorsten Wolff
Journal Submitted to: EMBO Molecular Medicine
Manuscript Number: EMM-2019-10938